# Complex opioid-driven modulation of glutamatergic and cholinergic neurotransmission in a GABAergic brain nucleus associated with emotion, reward, and addiction

Ramesh Chittajallu*, Anna Vlachos, Adam P Caccavano, Xiaoqing Yuan, Steven Hunt, Daniel Abebe, Edra London, Kenneth A Pelkey, Chris J McBain

Section on Cellular and Synaptic Physiology, Eunice Kennedy Shriver National Institute of Child Health and Human Development, National Institutes of Health, Bethesda, United States

*For correspondence:
ramesh.chittajallu@nih.gov

Competing interest: The authors declare that no competing interests exist.

## eLife Assessment

This study presents **important** information about the role of mu opioid receptors in neurotransmission between the medial habenula and the interpeduncular nucleus. The authors provide **convincing** evidence that mu opioid receptor activation has differential effects on transmission from substance P neurons and cholinergic neurons, and that blockade of potassium channels can unmask a nicotinic cholinergic synaptic response. This work will be of high interest to those studying this brain region, and potentially to the larger neuroscience community studying motivated behavior.

**Abstract** The medial habenula (mHb)/interpeduncular nucleus (IPN) circuitry is resident to divergent molecular, neurochemical, and cellular components which, in concert, perform computations to drive emotion, reward, and addiction behaviors. Although housing one of the most prominent mu-opioid receptor (mOR) expression levels in the brain, remarkably little is known as to how they impact mHb/IPN circuit function at the granular level. In this study, our systematic functional and pharmacogenetic analyses in mice demonstrate that mOR activation attenuates glutamatergic signaling while producing an opposing potentiation of glutamatergic/cholinergic co-transmission mediated by mHb substance P and cholinergic neurons, respectively. Intriguingly, this latter non-canonical augmentation is developmentally regulated only emerging during later postnatal stages. In addition, we reveal that specific potassium channels act as a molecular brake on nicotinic receptor signaling in the IPN with the opioid-mediated potentiation of this arm of neurotransmission being operational only following attenuation of Kv1 function. Thus, mORs play a complex role in shaping the salience of distinct afferent inputs and transmitter modalities that ultimately influence synaptic recruitment of downstream GABAergic IPN neurons. Together, these observations provide a framework for future investigations aimed at identifying the neural underpinnings of maladaptive behaviors that can emerge when opioids, including potent synthetic analogs such as fentanyl, modulate or hijack this circuitry during the vulnerable stages of adolescence and in adulthood.

## Introduction

Substance use disorders (SUDs) are agnostic to demographic group posing a widespread public health crisis impacting many communities (*Degenhardt et al., 2018*). Drugs of misuse influence synaptic function, resulting in acute and protracted circuit adaptations that can exacerbate compulsive behaviors resulting in dependence (*Lüscher and Malenka, 2011*; *Salmanzadeh et al., 2020*). Mu-opioid receptors (mORs) represent the primary molecular substrate responsible for both the clinically efficacious (i.e. pain management) and the euphoric effects promoting opioid misuse and addiction (*Matthes et al., 1996*). Additionally, mainly via liberation of endogenous opioids (*Trigo et al., 2010*), mORs also facilitate the addictive propensity of other substances (*Trigo et al., 2010*; *Darcq and Kieffer, 2018*; *Le Merrer et al., 2009*) for example alcohol and nicotine, which, together with opioids, comprise the three most prevalent drugs of abuse worldwide (*Degenhardt et al., 2018*). mORs are expressed by varied neural cell types resident in numerous regions within the brain (*Le Merrer et al., 2009*; *Reeves et al., 2022*; *Valentino and Volkow, 2018*). Thus, detailed investigations as to how mOR activation impacts circuit dynamics constitute an important endeavor critical for identification of potential therapeutic interventions aimed at alleviating the deleterious societal consequences of SUDs.

Here, we focus on a relatively understudied brain circuit comprising the medial habenula (mHb) and interpeduncular nucleus (IPN). The mHb is a bilateral epithalamic structure receiving significant synaptic input from, for example, the diagonal band and various septal regions (*Chung et al., 2023*; *Qin and Luo, 2009*; *Otsu et al., 2018*). The efferent output of the mHb is predominantly mediated by two distinct populations, substance P (SP) and cholinergic neurons, that primarily impinge on the IPN (*Qin and Luo, 2009*; *Ables et al., 2023*). This latter structure in the ventral midbrain is mainly populated with GABAergic neurons that influence activity in downstream brain regions such as the raphe nuclei and ventral tegmental area (*McLaughlin et al., 2017*). Thus, the mHb/IPN axis is anatomically positioned to integrate incoming limbic forebrain signals to ultimately control the function of these midbrain monoaminergic nuclei. It is therefore unsurprising that perturbation/modulation of the synaptic recruitment or activity of IPN GABAergic neurons precipitates varied emotion, reward, and addiction phenotypes (*Ables et al., 2023*; *McLaughlin et al., 2017*; *Ables et al., 2017*; *Molas et al., 2017*; *Klenowski et al., 2022*; *Tuesta et al., 2017*; *Zhao-Shea et al., 2013*; *Wolfman et al., 2018*; *Souter et al., 2022*; *Liang et al., 2024*).

The mHb/IPN axes house one of the highest densities of mORs in the central nervous system (*Gardon et al., 2014*; *Bailly et al., 2020*), thus representing a prominent brain locus for opioid action. Despite this, descriptions regarding the role of mHb/IPN mORs in shaping behavior are still in their relative infancy (*Allain et al., 2022*; *Boulos et al., 2020*). mOR activation and subsequent G-protein signaling (Gi/o) classically produces an acute direct inhibitory effect in many neuronal types typified by membrane potential hyperpolarization via activation of inward rectifying potassium (GIRK) channels and/or reduced function of voltage-gated calcium channels that are coupled to neurotransmitter release (*Reeves et al., 2022*; *Law et al., 2000*). Strikingly, there is a dearth of information concerning the cellular effects and subsequent circuit influence of mOR signaling in the mHb/IPN, rendering the neural correlates underlying the behavioral outcomes elicited by opioids unclear at present. Interestingly, direct excitatory effects of mOR activation are apparent in certain areas of the nervous system either under basal conditions or following chronic drug regimens (*Margolis et al., 2014*; *Hashimoto et al., 2009*; *Crain and Shen, 1990*; *Madhavan et al., 2010*). Furthermore, within the IPN itself, GABA$_B$-receptor activation (another Gi-linked receptor) produces an increase in synaptic transmission (*Koppensteiner et al., 2024*; *Zhang et al., 2016*; *Bhandari et al., 2021*), questioning whether this surprising effect is generalized to other members of this receptor family expressed within this region such as mORs.

The central role of the mHb/IPN circuitry underlying numerous behavioral aspects relating to emotion and the cycle of addiction, the conspicuous expression of mORs and a clear precedent for non-canonical effects of Gi-linked receptors together formed the impetus for the current work. Employing cell-type conditional optogenetics in combination with slice electrophysiology and pharmacogenetics, we systematically dissect the effects of opioids on the mHb/IPN circuitry with a particular focus on synaptic signaling between these structures.

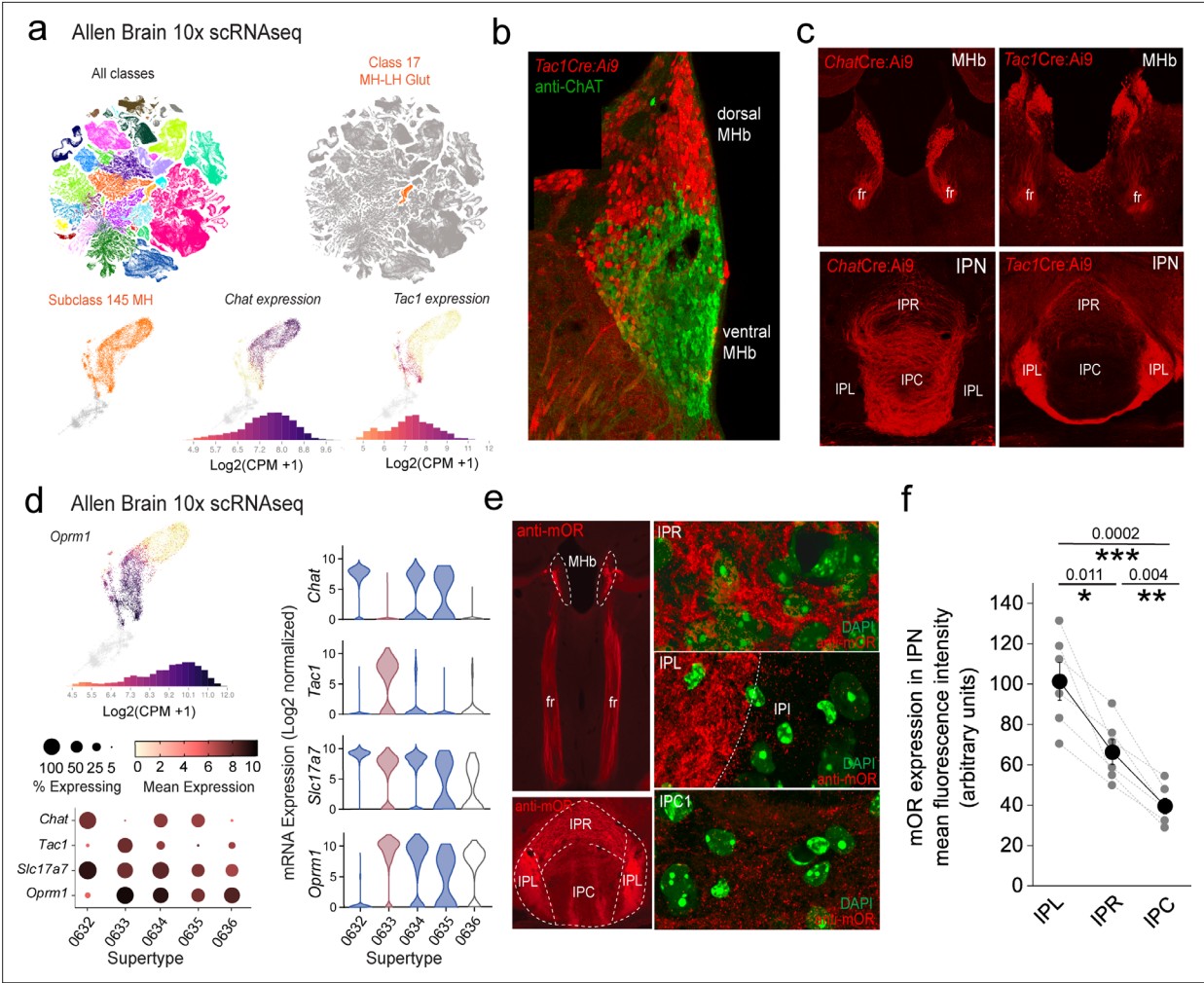

**Figure 1.** *Oprm1* gene and mu-opioid receptor (mOR) protein expression in the habenula–interpeduncular axis. (**a**) 10X scRNAseq UMAP whole mouse brain showing all cellular classes (top left panel; 4.04 million cells) and Class 17 MH-LH Glut (right panel; corresponding to the medial habenula (mHb) and lateral habenula; 10.8K cells). Bottom panels focus on individual mHb cells (Subclass 145 MH; 8K cells) indicating log$_2$ mRNA expression of *Chat* and *Tac1*. (**b**) Confocal image of a *Tac1*Cre:Ai9 mouse immunostained with anti-ChAT illustrating the distinct distribution of substance P (red) and cholinergic neurons (green) in dorsal and ventral mHb, respectively. (**c**) Conditional td-Tomato expression in cholinergic (left panels; *Chat*Cre:Ai9) and SP neurons (right panels; *Tac1*Cre:Ai9) illustrating their spatial location within the mHb with their axonal outputs in the fasciculus retroflexus (fr) and largely non-overlapping terminal axonal arborization patterns in interpeduncular nucleus (IPN). (**d**) Profile of log$_2$ mRNA expression of *Oprm1* in Subclass 145 MH (top left panel). Corresponding dot and violin plots depicting three *Chat* (0632, 0634, and 0635) and one *Tac1* (0633) supertype with relative expression of *Slc17a7* (VGluT1) and *Oprm1* in each corresponding supertype. In the violin plots, the *Chat* and *Tac1* supertypes are color-coded blue and red, respectively. (**e**) Endogenous mOR expression throughout the mHb and IPN axis (left panels; red) assessed by immunocytochemistry. High-resolution airy scan images of mOR distribution in subdivisions of the IPN; rostral IPN (IPR), lateral IPN (IPC), and central IPN (IPC). Green fluorescence is DAPI-stained nuclei. (**f**) Densitometry analyses of mOR expression in the various subfields of IPN. Data are from two to four slices containing IPN taken from each of six mice aged P40–P60. Data depicted in (**a**) and (**d**) are from the publicly available Allen Brain cell Atlas (https://knowledge.brain-map.org/abcatlas). See methods for further details.

## Results

The two major neuronal populations of the mHb, substance P (SP) and cholinergic subtypes, can be distinguished by virtue of *Tac1* and *Chat* gene expression. Leveraging the Allen Brain Institute's publicly available whole mouse brain 10X single-cell RNAseq dataset (https://alleninstitute.github.io/abc_atlas_access/intro.html) (*Yao et al., 2023*), it is evident that these two neuronal classes within the mHb cluster (Subclass 145 MH) can be parsed based on their transcriptomic profiles (*Figure 1a*). Spatially, the SP and cholinergic neurons largely segregate to the dorsal versus ventral mHb, respectively (*Figure 1b, c*). The projection pattern as delineated by use of specific Cre transgenic mouse

lines (i.e *Chat*Cre and *Tac1*Cre) when crossed to a conditional TdTomato reporter (Ai9) reveals a predominant innervation of IPL by SP neurons, whereas cholinergic neurons impinge on the IPR/IPC subdivisions (*Figure 1c*). Together, these distinct neuronal classes provide much of the afferent input to the IPN via the fasciculus retroflexus (fr) axonal tracts (*Figure 1c*) ultimately dictating IPN neuronal recruitment. At the mRNA level, two of the designated three *Chat* neuronal Supertypes and the single *Tac1* Supertype demonstrate significant expression of *Oprm1* (mean expression levels including zero values = 0.33, 6.8, and 4.1 in *Chat* Supertypes 0632, 0634, and 0635, respectively, and 8.7 in the *Tac1* Supertype 0633; *Figure 1d*). At the protein level, and as previously reported (*Gardon et al., 2014*), we confirm that mOR expression is notable in the mHb, the fasciculus retroflexus axonal tracts, and particularly prevalent in the IPN (*Figure 1e*). Within this latter structure, high-resolution microscopy clearly shows mOR expression is densest in the lateral IPN (IPL) with intermediate and relatively lower levels found in the rostral IPN (IPR) and central IPN (IPC) subregions, respectively (*Figure 1e, f*).

Selective stimulation of each of these mHb neuronal populations in isolation is achieved using Cre-mediated conditional expression of channel rhodopsin by crossing Ai32 mice with either *Chat*Cre or *Tac1*Cre mice; see methods for details. Adopting this optogenetic approach, we investigate how mOR receptor activation modulates habenulo-interpeduncular synaptic dialog imparted by these parallel yet distinct afferent systems in adult mice (p > 40). Both SP and cholinergic neurons express the glutamate vesicular transporter, VGluT1 (*Slc17a7*; *Figure 1d*) and competently release this excitatory neurotransmitter (*Souter et al., 2022*; *Ren et al., 2011*; *Singhal et al., 2025*).

## mOR activation reduces substance P neuronal-mediated glutamatergic transmission onto IPL GABAergic neurons

Since the highest expression of mOR was found in the IPL (*Figure 1e, f*), we initially probed the effect of a selective mOR agonist (DAMGO, 500 nM) on AMPA receptor-mediated transmission mediated by SP neurons that prominently innervate this subregion (*Figure 1c*). DAMGO application results in a significant decrease in light-driven (470 nM; typically, 10–50% arbitrary LED power equating to approximately 0.4–3.4 mW/mm², CoolLED illumination system) AMPAR-mediated EPSCs in *Tac1*Cre:Ai32 mice that is partially reversed upon washout of the agonist (*Figure 2a, b*). This is accompanied by an increase in S2/S1 paired pulse ratio (PPR) suggesting the modulation is driven by presynaptic changes in release probability (*Figure 2c*). Thus, neurotransmitter release by this specific afferent input to the IPL is directly, negatively modulated like that seen in many other neural circuits within the brain (*Reeves et al., 2022*).

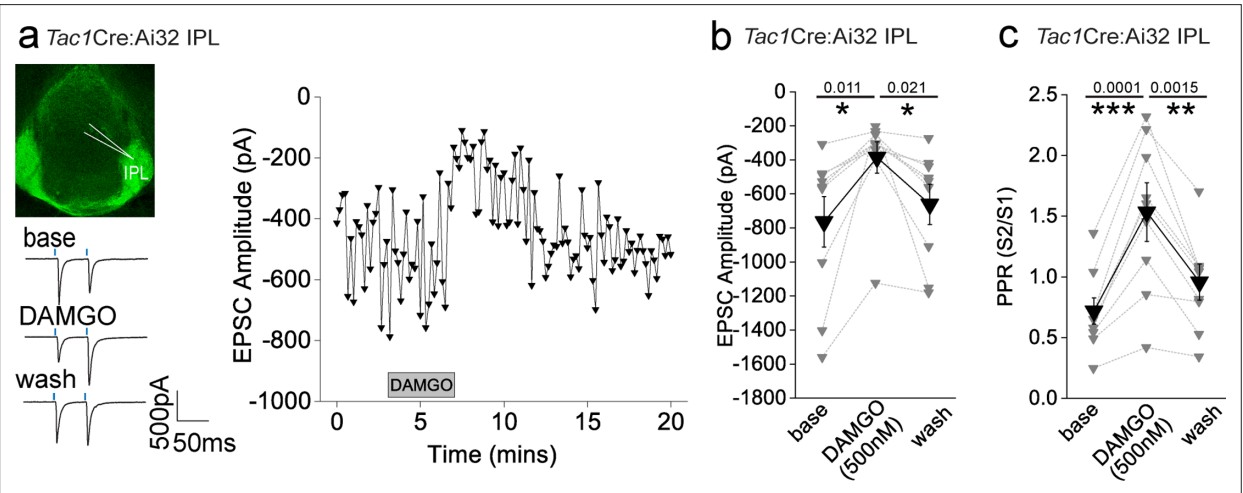

**Figure 2.** AMPAR-mediated synaptic transmission in lateral IPN (IPL) mediated by substance P neurons is inhibited by mu-opioid receptor (mOR) activation. (**a**) Whole-cell voltage-clamp in adult (>p40) *Tac1*Cre:Ai32 mice (top left panel illustrating the axonal arborization of ChR2 expressing SP neuronal axons in IPN and the position of neuronal recording in IPL). Single voltage-clamp traces of light-evoked AMPAR EPSCs (bottom left panel; 470 nM light pulse; two stimulations at 20 Hz, blue dashes) and time course of peak amplitudes (right panel) under baseline, during application of 500 nM DAMGO and washout conditions. (**b, c**) Individual (gray filled symbols) and mean (black filled symbols) data of AMPAR EPSC amplitude and corresponding paired pulse ratios (PPRs; *n* = 9 recorded IPL neurons from 7 mice).

# mOR activation augments the strength and fidelity of glutamatergic transmission mediated by cholinergic neurons, resulting in enhanced excitation-spike coupling in IPR GABAergic neurons

We next examined how mORs modulate mHb-IPN neurotransmission that occurs via the other major input mediated by cholinergic neurons. To this end, we employed *Chat*Cre:Ai32 mice and focused on the IPR (*Figure 3a*) due to the relatively high levels of combined mOR expression and cholinergic neuronal innervation (*Figure 1c, e, f*) in this subregion. Under our basal experimental conditions and in agreement with previous studies (*Souter et al., 2022*; *Singhal et al., 2025*; *Ren et al., 2022*), light-evoked postsynaptic responses in the IPN mediated by mHb cholinergic neurons in response to either brief single or paired pulse stimulation are solely mediated by AMPARs as evidenced by complete pharmacological block by DNQX (refer to *Figure 5—figure supplement 1a* and *Figure 7— figure supplement 1a–c*). Remarkably, DAMGO application elicited a significant robust and reversible potentiation of light-evoked (470 nM; typically, 10–100% arbitrary LED power equating to approximately 0.4–6.9 mW/mm$^2$; CoolLED illumination system) AMPAR EPSC amplitude (*Figure 3a, b*) in stark contrast to that seen with mHb *Tac1*-IPL glutamatergic signaling described previously (*Figure 2a, b*). Notably, the accompanying PPR changes were not consistent with an increase in release probability (*Figure 3c*), raising uncertainty as to the synaptic locus of the mOR effect.

In these experiments, we were 'blind' to the molecular identity of the postsynaptic IPR neuron. Since a major population of neurons in this IPN subregion is of the somatostatin (SST) GABAergic subtype (*Ables et al., 2017*; *Hsu et al., 2013*), it is likely that most of our recordings are from this subtype. Nevertheless, we also employed *Chat*-ChR2 transgenic mice containing fluorescently reported SST neurons (i.e. *Chat*-ChR2:SSTCre:Ai9; *Figure 3d*) to allow targeted recordings specifically from these neurons. Our data clearly show that DAMGO robustly increases AMPAR EPSCs impinging on IPR RFP+ SST neurons (*Figure 3d, e*) accompanied by a trending increase in PPR (*Figure 3f*). We also performed a series of additional experiments agnostic to the postsynaptic neuronal identity in IPC subdivision. Despite the relatively low expression levels of mOR in this subdivision of IPN (*Figure 1e, f*), we observed a robust increase in AMPAR EPSC amplitude at a similar extent to that seen in IPR (*Figure 3—figure supplement 1*), thus extending this remarkable potentiation of glutamatergic synaptic transmission to all IPN subfields where cholinergic neuronal terminals reside.

We considered non-specific actions of DAMGO such as those that could be mediated by a possible direct modulation of ChR2 itself to explain this non-canonical observation. However, our previous data demonstrating the reduction of glutamatergic neurotransmission mediated by SP neurons by DAMGO described (*Figure 2*) renders this possibility unlikely. Furthermore, a similar potentiation of cholinergic neuronal-mediated glutamatergic transmission in the IPN upon activation of GABA$_B$ receptors has demonstrated a propensity for this circuit to undergo such modulation in response to a Gi-linked receptor (*Koppensteiner et al., 2024*; *Zhang et al., 2016*; *Bhandari et al., 2021*). Nevertheless, despite this precedent, we employed a pharmacological approach to further validate this unexpected result. We show that lower concentrations of DAMGO (100 nM) and alternative mOR agonists, Met-enkephalin (Met-enk, 3 µM) and morphine (10 µM), all induce potentiation of ChAT neuronal glutamatergic transmission to similar extents (*Figure 3g*). In addition, pre-treatment with the mOR selective antagonist (CTAP, 1 µM) completely prevents the DAMGO-induced response (*Figure 3g*). Together, the pharmacological battery of tests employing an experimental, an endogenous, and clinical/recreational mOR agonists and selective mOR antagonism clearly implicate this opioid receptor in mediating a non-canonical potentiation of AMPAR EPSC amplitude elicited by mHb cholinergic neurons in IPN.

In several recordings from both 'blind' patching of IPR neurons and in directly identified SST neurons, no measurable postsynaptic AMPA receptor EPSC could be elicited following light-evoked stimulation of cholinergic axons even with the maximal possible light intensity available (i.e. 100% arbitrary LED power; approximately 6.9 mW/mm$^2$), and thus further interrogation was typically not performed. However, in a few instances, we nevertheless proceeded with DAMGO or Met-enk application, and surprisingly, in a subset of these recordings (9/15 cells tested), a robust and reversible emergence of a significant AMPAR-mediated EPSC was observed (*Figure 3h, i*). Thus, these data clearly reveal that mOR activation invokes a synaptic mechanism that, at its most extreme, results in an 'un-silencing' of ChAT neuronal glutamatergic transmission in IPN including in directly identified SST GABAergic neurons. Since PPR measurements are only valid if the same population of release sites

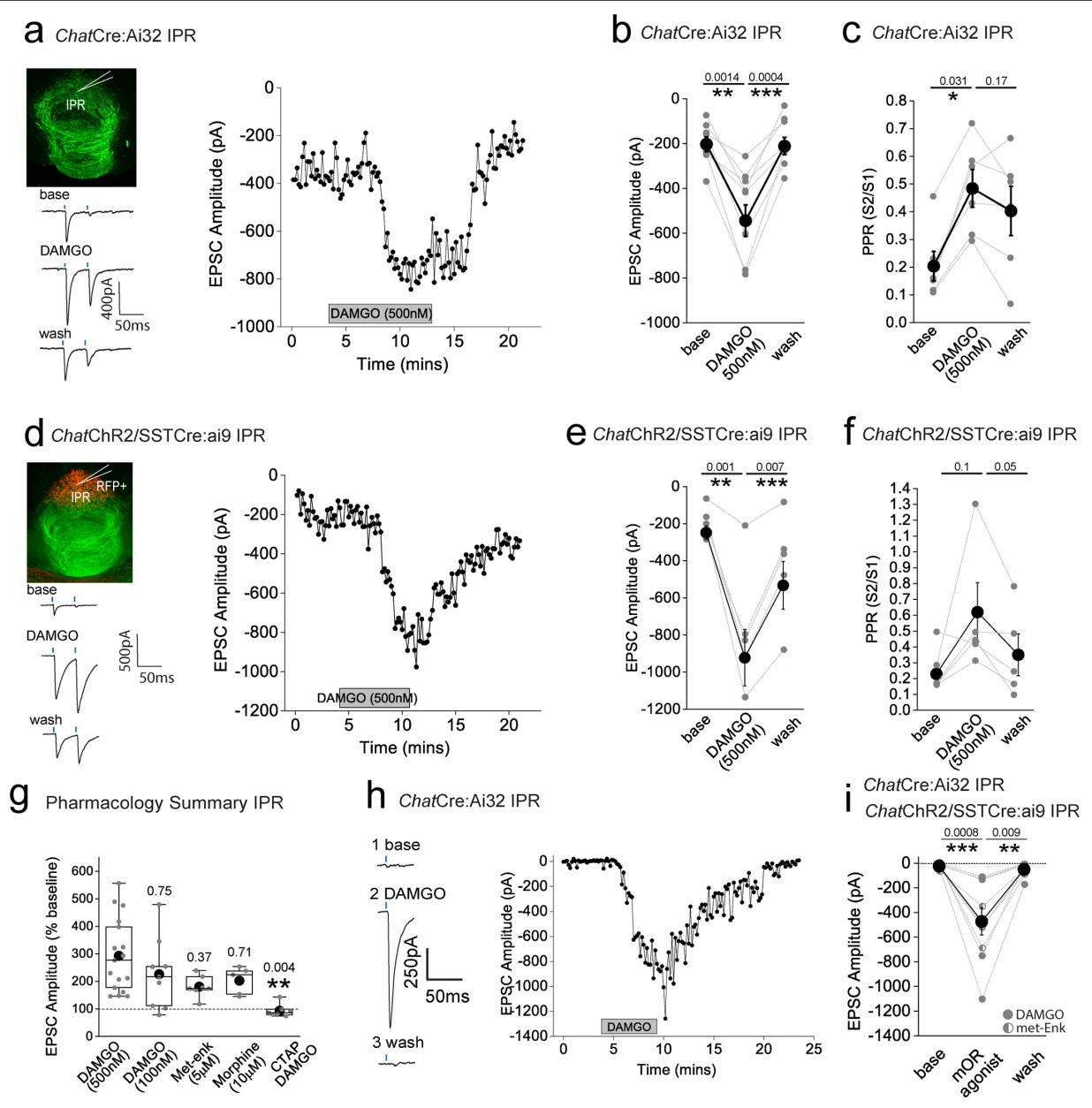

**Figure 3.** AMPAR-mediated synaptic transmission in rostral IPN (IPR) mediated by medial habenula (mHb) cholinergic neurons is potentiated by mu-opioid receptor (mOR) activation. (**a**) Whole-cell voltage-clamp in adult (>p40) *Chat*Cre:Ai32 mice (top left panel illustrating the axonal arborization of ChR2 expressing cholinergic neuronal axons in IPN and the position of neuronal recording in IPR). Single voltage-clamp traces of light-evoked AMPAR EPSCs (bottom left panel; 470 nM light pulse; two stimulations at 20 Hz, blue dashes) and time course of peak amplitudes (right panel) under baseline, during application of 500 nM DAMGO and washout conditions. (**b, c**) Individual (gray filled symbols) and mean (black filled symbols) data of AMPAR EPSC amplitude and corresponding paired pulse ratios (PPRs; *n* = 8 and 6 recorded IPR neurons from 7 and 5 mice for AMPAR EPSC amplitude and PPR, respectively). (**d**) Whole-cell voltage-clamp in adult (>p40) *Chat*ChR2:SSTCre:Ai9 mice (top left panel illustrating the axonal arborization of ChR2 expressing cholinergic neuronal axons in IPN and the position of neuronal recording in RFP+ SST IPR neurons). Single voltage-clamp traces of light-evoked AMPAR EPSCs (bottom left panel; 470 nM light pulse; two stimulations at 20 Hz, blue dashes) and time course of peak amplitudes (right panel) under baseline, during application of 500 nM DAMGO and washout conditions. (**e, f**) Individual (gray filled symbols) and mean (black filled symbols) data of AMPAR EPSC amplitude and corresponding PPRs (*n* = 5 recorded IPR neurons from 4 mice). (**g**) Summary bar graph illustrating the effect of DAMGO (500 and 100 nM; *n* = 16 recorded IPR neurons from 14 mice and *n* = 9 recorded IPR neurons from 4 mice, respectively), met-enkephalin (5 μm; *n* = 7 recorded IPR neurons from 5 mice), morphine (10 μM; *n* = 5 recorded IPR neurons from 5 mice) and 500 nM DAMGO in presence of mOR antagonist 1 mM CTAP (*n* = 7 recorded IPR neurons from 2 mice). (**h**) Voltage-clamp trace examples (single light stimulus; blue bar; left panel) and time course (right panel) under baseline, DAMGO, and washout conditions in a recorded IPR neuron displaying no measurable baseline AMPAR EPSC in response to maximal light-evoked stimulation (470 nm; 6.9 mW/mm²). (**i**) Individual (gray filled and half-filled symbols for 500 nM DAMGO or 5 μM Met-enkephalin,

*Figure 3 continued on next page*

*Figure 3 continued*

respectively) and pooled (black filled symbols) data of light-evoked AMPAR EPSC amplitude during DAMGO application and washout (n = 6 and 3 recorded IPR cells from 6 and 3 mice for DAMGO and met-enkephalin application, respectively).

The online version of this article includes the following figure supplement(s) for figure 3:

**Figure supplement 1.** AMPAR-mediated synaptic transmission in central IPN (IPC) mediated by medial habenula (mHb) cholinergic neurons is potentiated by mu-opioid receptor (mOR) activation.

are assayed before and after experimental manipulation, the additional engagement of a putative reluctant vesicular pool (*Koppensteiner et al., 2024*) by mOR activation could explain the discrepancy between the augmentation of EPSC amplitude and observed changes in PPR (*Figure 3b, c, e, f*). Future interrogation is warranted for definitive identification of the relative contributions of pre- and postsynaptic loci to the potentiation observed (but see *Singhal et al., 2025*).

To date, we have examined the effect of mOR on synaptic transmission during paired pulse light activation delivered at 20 Hz. Although mHb cholinergic neurons are capable of burst firing at much

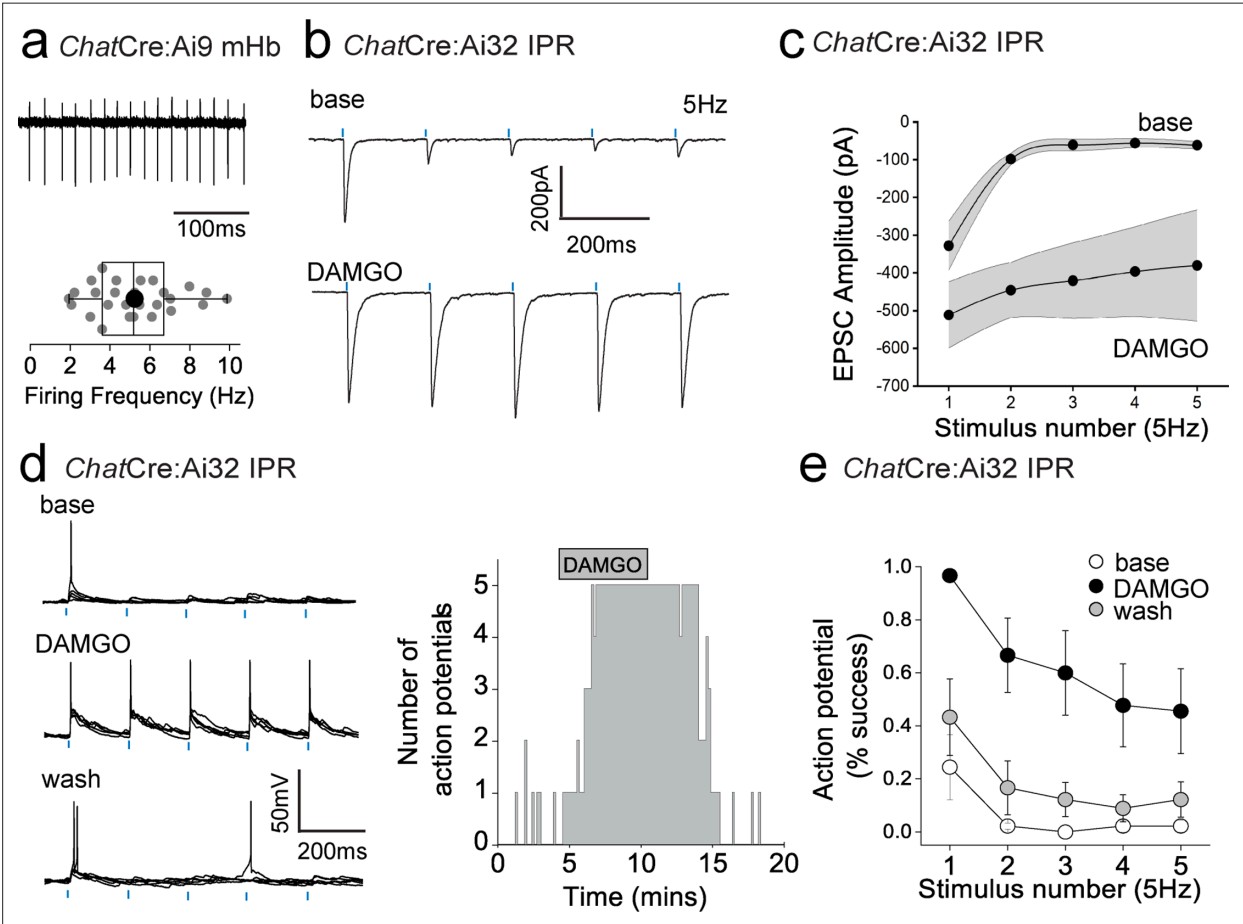

**Figure 4.** Mu-opioid receptor (mOR) activation increases fidelity of glutamatergic transmission mediated by medial habenula (mHb) cholinergic neurons to augment excitation:spike coupling in postsynaptic IPR neurons. (**a**) Single example trace showing spontaneous action potential firing in cell-attached mode from a td-Tomato-positive ventral mHb neuron in the *Chat*Cre:Ai9 mouse (top panel). Box plot and corresponding individual data of the spontaneous firing frequency of mHb cholinergic neurons (bottom panel; n = 27 recorded cells). (**b**) Single voltage-clamp traces of light-evoked (470 nm, five pulses delivered at 5 Hz; blue dashes) AMPAR EPSCs recorded from postsynaptic IPR neurons in adult *Chat*Cre:Ai32 mice (p > 40) under baseline and following 500 nM DAMGO application (left panel). (**c**) Pooled data (shaded area denotes SEM) of peak amplitude for each given stimulus in the 5 Hz train during baseline and following 500 nM DAMGO application (n = 5 recorded IPR neurons from 5 mice). (**d**) Single current-clamp traces example (five consecutive overlaid sweeps) of light-evoked EPSP:action potential coupling (five stimuli at 5 Hz; blue dashes) under baseline, 500 nM DAMGO, and washout conditions (left panel). Corresponding single example time course plot depicting number of light-driven EPSP-evoked action potential (right panel). (**e**) Mean data showing percentage success over five consecutive traces of EPSP:action potential coupling for each given stimulus in the 5 Hz train under baseline (open symbols), 500 nM DAMGO (black symbols), and washout (gray symbols) conditions (n = 9 recorded IPR neurons from 7 mice).

higher frequencies in response to afferent stimulation in vivo (*Otsu et al., 2018*), they typically exhibit low frequency (i.e. <10 Hz) intrinsic (i.e. independent of synaptic input) firing in vitro (*Chung et al., 2023*; *Arvin et al., 2019*; *Görlich et al., 2013*; *Cho et al., 2020*). In agreement with these previous studies, cholinergic neurons in ventral mHb (identified in ChatCre:Ai9 mice; *Figure 1c*) spontaneously elicit action potentials measured in cell-attached recordings at an average of ~5 Hz (range 2–10 Hz; *Figure 4a*). Therefore, we assessed the role of mOR activation on glutamatergic transmission mediated by cholinergic neurons elicited by light stimulation trains delivered at this frequency. Interestingly, even with this relatively low frequency paradigm, glutamatergic transmission is extremely labile in nature as evidenced by the large and rapid depression of AMPAR EPSC during the stimulus train (*Figure 4b, c*). Remarkably, activation of mORs essentially eliminates activity-dependent depression at this synapse (*Figure 4b, c*). This switch in transmission dynamics greatly facilitates the probability of excitatory postsynaptic potential-spike coupling (ES coupling) in response to each stimulus within the train (*Figure 4d, e*). Thus, these data clearly demonstrate that at physiologically relevant stimulus patterns, mORs serve to dramatically increase the fidelity of glutamatergic transmission in the IPR mediated by cholinergic neurons culminating in an enhanced recruitment of postsynaptic IPR neurons.

## mORs constitute a molecular switch to alter the salience of glutamatergic transmission mediated by cholinergic and substance P neurons in the IPR

An intriguing inconsistency was observed during the conduction of our experiments regarding the DAMGO effect on light-evoked and spontaneous EPSCs (sEPSCs) impinging on IPR neurons. Under the experimental conditions employed, sEPSC events were exclusively mediated by AMPARs as evidenced by their virtually complete cessation upon DNQX application (*Figure 5—figure supplement 1a*). Remarkably, in contrast to the previously described robust potentiation of light-evoked AMPAR EPSCs, activation of mORs significantly attenuates sEPSC frequency in agreement with a recent study (*Singhal et al., 2025*), with no effect on amplitude (*Figure 5—figure supplement 1b–d*). Thus, in a single IPR postsynaptic neuron, opposing effects on light-evoked and spontaneous AMPAR EPSCs are evident (*Figure 5—figure supplement 1e*). Although these data do not directly identify the origin and neuronal subtype(s) responsible for the sEPSCs measured, the incongruent effect of mOR activation on evoked versus spontaneous events led us to consider the possible existence of an additional afferent system impinging on IPR neurons. Prevailing circuit schemas based on axonal arborization patterns portray a mutually exclusive, non-overlapping afferent input to IPN mediated by cholinergic and SP neurons to IPR/IPC versus IPL, respectively. However, in the current study by employing high-resolution imaging in *Tac1*Cre:Ai32 mice, a clear presence of synaptic bouton-like structures in the IPR is noted (*Figure 5a, b*). These structures do not co-localize with endogenous ChAT, but do express the glutamate vesicular transporter, VGluT1 (*Figure 5a, b*). This presence of putative presynaptic anatomical substrates points to the existence of a secondary input to IPR mediated by SP neurons distinct to the well-established one originating from mHb cholinergic neurons. Indeed, significant light-evoked AMPAR EPSCs in IPR neurons can be elicited in *Tac1*Cre:Ai32 mice with essentially similar amplitudes across all LED powers employed to that seen in *Chat*Cre:Ai32 mice (*Figure 5c*). Thus, in contrast to recent ultrastructural EM analyses (*Koppensteiner et al., 2024*), these data indicate a previously unidentified functional glutamatergic input to IPR mediated by SP neurons. Interestingly, DAMGO application significantly depresses the SP neuronal glutamatergic output in IPR and increases PPR (*Figure 5d–f*) to a similar extent to that observed in IPL (*Figure 2*). This is in direct contrast to the role of mORs in positively modulating the ChAT neuronal glutamatergic transmission in the IPR (*Figure 3a, b, d, e*). Thus, for the first time, we demonstrate both SP and cholinergic input to the same subregion of IPN and that mOR activation elicits diametrically opposing effects on transmission mediated by these respective presynaptic neuronal populations.

## Dynamic regulation of the mOR elicited afferent specific switch in salience of the glutamatergic transmission to the IPR during adolescence

Our functional studies thus far have been restricted to sexually mature, adult mice (>p40). mOR expression emerges in many brain structures during prenatal development largely overlapping with their final adult distribution profile by mid/late gestation (*Zhu et al., 1998*). In agreement, we reveal that

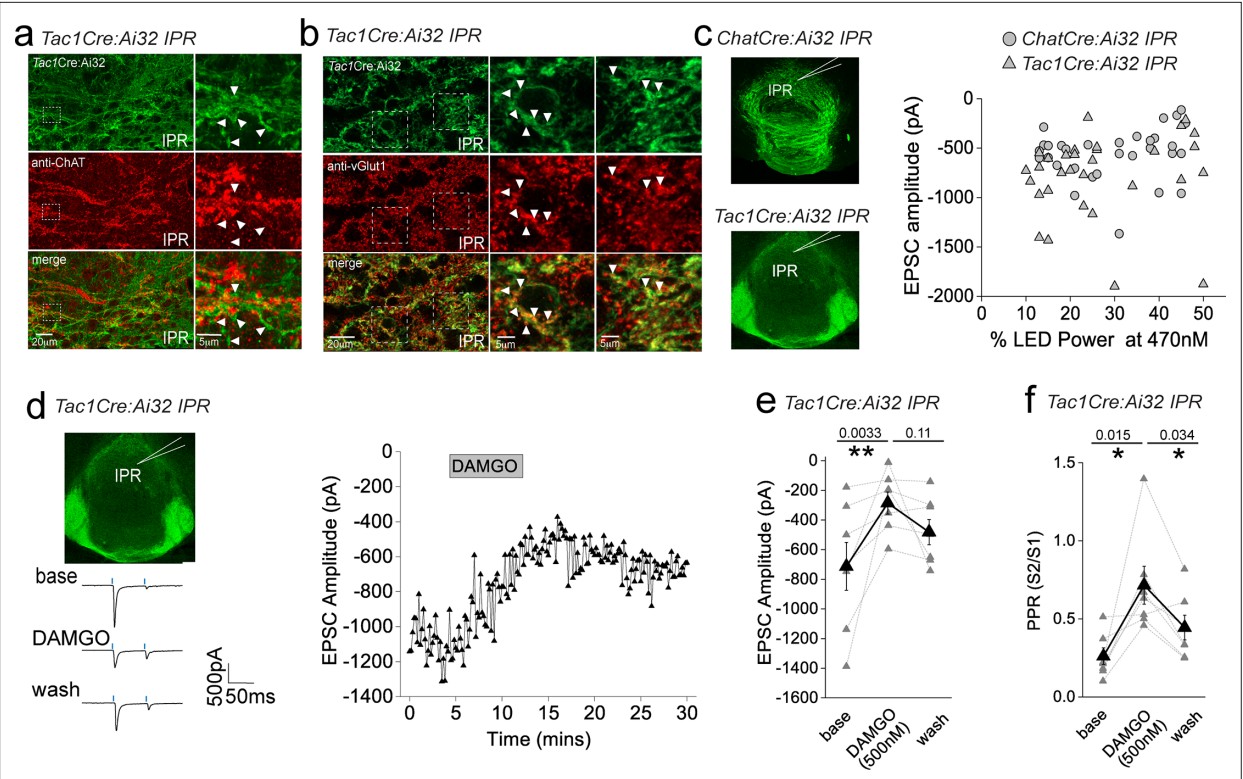

**Figure 5.** Effect of mu-opioid receptor (mOR) activation on a novel functional SP neuronal-mediated evoked AMPAR EPSCs in IPR. (**a**) High-resolution airy scan of the IPR in *Tac1*Cre:Ai32 mouse showing ChR2-expressing synaptic bouton-like structures (green) and endogenous ChAT expression (red). Right panels are magnified regions of the boxed area. Arrows indicate examples of non-overlap of TacCre:Ai32 boutons with ChAT. (**b**) High-resolution airy scan of the IPR in *Tac1*Cre:Ai32 mouse showing ChR2-expressing synaptic bouton-like structures (green) and endogenous VGlut1 expression via immunostaining (red). Right panels are magnified regions of the boxed area. Arrows indicate examples of expression of VGluT1 within TacCre:Ai32 bouton structures. (**c**) Comparison of light-evoked AMPAR EPSC peak amplitude in postsynaptic IPR neurons mediated by either cholinergic (*Chat*Cre:Ai32 mice; n = 32 recorded IPR neurons) or substance P (*Tac1*Cre:Ai32 mice; n = 31 recorded IPR neurons) medial habenula (mHb) neurons at various arbitrary % LED (10–50% corresponding to an approximate power of 0.4–3.4 mW/mm²; right panel). (**d**) Whole-cell voltage-clamp in adult (>p40) *Tac1*Cre:Ai32 mice (top left panel illustrating the axonal arborization of ChR2-expressing SP neuronal axons in IPN and the position of neuronal recording in IPR). Single voltage-clamp traces of light-evoked AMPAR EPSCs (bottom left panel; 470 nM light pulse; two stimulations at 20 Hz, blue dashes) and time course of peak amplitudes (right panel) under baseline, during application of 500 nM DAMGO and washout conditions (right panel). (**e, f**) Individual (gray filled symbols) and mean (black filled symbols) data of AMPAR EPSC amplitude and corresponding paired pulse ratios (PPRs; n = 8 recorded IPR neurons from 7 mice).

The online version of this article includes the following figure supplement(s) for figure 5:

**Figure supplement 1.** Mu-opioid receptor (mOR) activation imparts opposing effects on spontaneous AMPA EPSCs versus cholinergic neuron-mediated evoked AMPAR EPSCs in IPR.

mOR protein expression in IPR is present at appreciable levels from early postnatal stages (i.e. from P10 onwards; *Figure 6a*). This begs the question of whether modulation of habenulo-interpeduncular synaptic transmission described during these critical early epochs mirrors that seen in the adult. To examine this, we extended our analyses to earlier time points (P15–P40) and found that mOR modulation of glutamatergic transmission mediated by cholinergic neurons in the IPR undergoes a remarkable developmental regulation. This is characterized by an initial inhibitory effect on light-evoked AMPAR EPSCs transitioning to the previously described augmentation (*Figure 6b, c, f*). In contrast, the glutamatergic input to the IPL and the newly discovered input to the IPR elicited by SP neurons is not developmentally regulated with synaptic transmission being consistently inhibited by DAMGO application at all ages tested (*Figure 6d–f*). Thus, despite essentially similar levels of IPR mOR expression at the ages assayed (*Figure 6a*), at earlier development stages, opioids result in overall inhibition of afferent input to the IPR agnostic to afferent input with the emergence of the previously highlighted differential regulation of mHb cholinergic versus substance P neuronal output emerging during the late adolescent stages (*Figure 6f*).

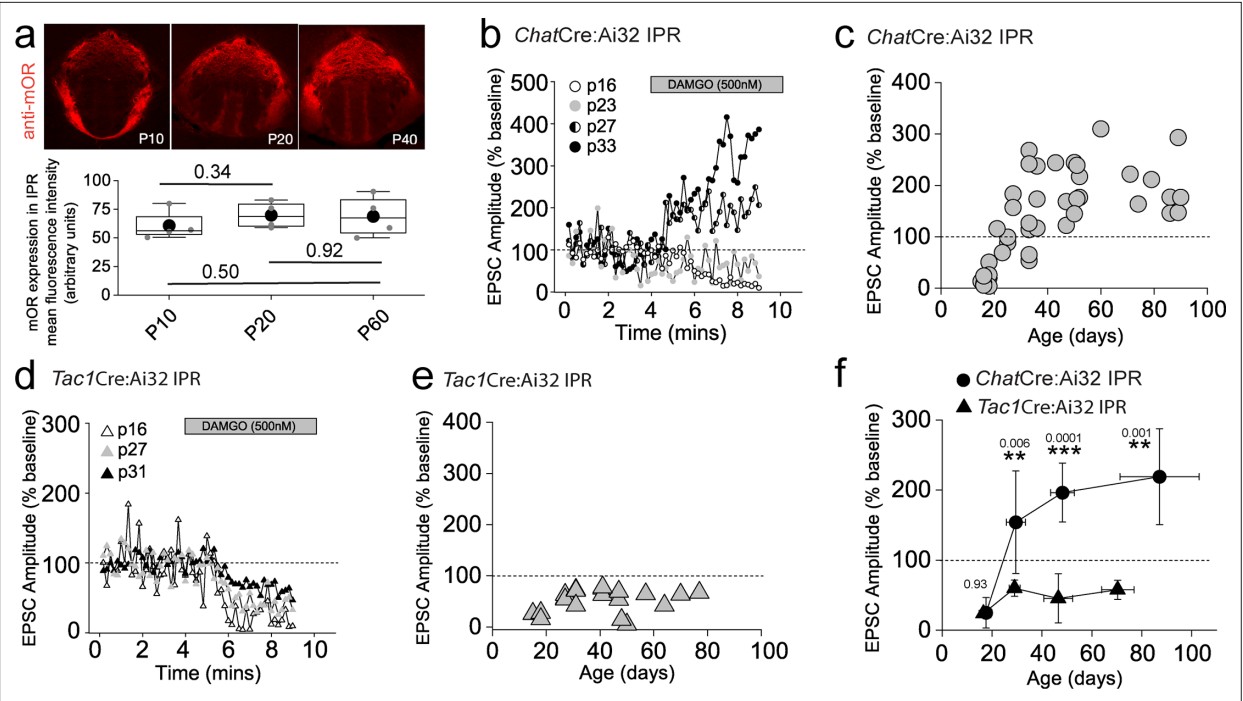

**Figure 6.** Mu-opioid receptors (mORs) constitute a developmentally regulated molecular switch altering the salience of neurotransmission in IPR mediated by substance P versus cholinergic neurons. (**a**) Confocal images of mOR protein expression in IPN during development (p10, p20, and p40). Densitometry analyses of mOR protein expression in IPR across development measurements taken from two slices containing IPR from each of two to four mice for each age. (**b**) Single examples of the time course of light-evoked AMPAR EPSC peak amplitudes mediated by medial habenula (mHb) cholinergic neurons following DAMGO application in postsynaptic IPR neurons at varying ages as indicated. (**c**) Individual data of the percent change of light-evoked cholinergic neuronal-mediated AMPAR EPSC peak amplitude elicited following DAMGO application across all ages tested (*n* = 44 recorded IPR neurons). (**d**) Single examples of the time course of light-evoked AMPAR EPSC peak amplitudes mediated by mHb substance P neurons following DAMGO application in postsynaptic IPR neurons at varying ages as indicated. (**e**) Individual data of the percent change of light-evoked SP neuronal-mediated EPSC peak amplitude elicited following DAMGO application across all ages tested. (**f**) Summary plot of the mean changes in the normalized AMPAR EPSC peak amplitude (% baseline) by DAMGO mediated by cholinergic and SP neurons binned at the following developmental epochs; p15–23 (postnatal), p24–34 (adolescent/pre-pubescent), p35–60 (adolescent/pubescent, sexual maturation) and >p60 (adult) (*Brust et al., 2015*). Total numbers of cells recorded = 39 and 21 cells from *Chat*Cre:Ai32 and *Tac1*Cre:Ai32 mice, respectively. Error bars denote standard deviation of the mean. Note datapoints for ages >p40 in panels **c** and **e** are normalized data taken from recorded cells that were depicted in *Figure 3b* and *Figure 5e* as absolute peak amplitude changes, respectively.

## mOR-induced potentiation of nicotinic receptor signaling in IPR is conditional on the removal of a molecular brake mediated by Kv1 channel function

In addition to mORs, this circuitry contains dense expression of nicotinic receptors located at pre- and postsynaptic sites (*Chung et al., 2023*; *McGehee et al., 1995*; *Shih et al., 2014*). However, the ability to reliably evoke postsynaptic nicotinic receptor (nAChR) EPSCs in response to physiologically appropriate stimuli has been challenging. Indeed, many studies have resorted to high frequency and prolonged stimulation (*Ren et al., 2011*) or bath/puff applications of nicotine to investigate nAChR function in this circuit (*Zhao-Shea et al., 2013*; *Hsu et al., 2013*; *Arvin et al., 2019*; *Wei et al., 2022*; *Elayouby et al., 2021*; *Mondoloni et al., 2023*). Despite the faithful co-expression of endogenous ChAT in *Chat*Cre:Ai32 expressing terminals located in the IPR (*Figure 7a*), evoked transmission by brief light pulses (1–5 ms) from cholinergic neurons predominantly elicits pure AMPAR EPSC in post-synaptic IPR neurons under our basal conditions as described above and in agreement with previous observations (*Souter et al., 2022*; *Ren et al., 2011*; *Singhal et al., 2025*; *Hsu et al., 2013*) (see *Figure 5—figure supplement 1a* and *Figure 7—figure supplement 1a-e*). Additionally, DAMGO, at concentrations that strongly potentiate glutamatergic transmission, does not result in a measurable postsynaptic nAChR EPSC after AMPARs are pharmacologically silenced (*Figure 7—figure supplement 1a–c*). Furthermore, repeated light stimulation at either 25 or 50 Hz fails to elicit the emergence

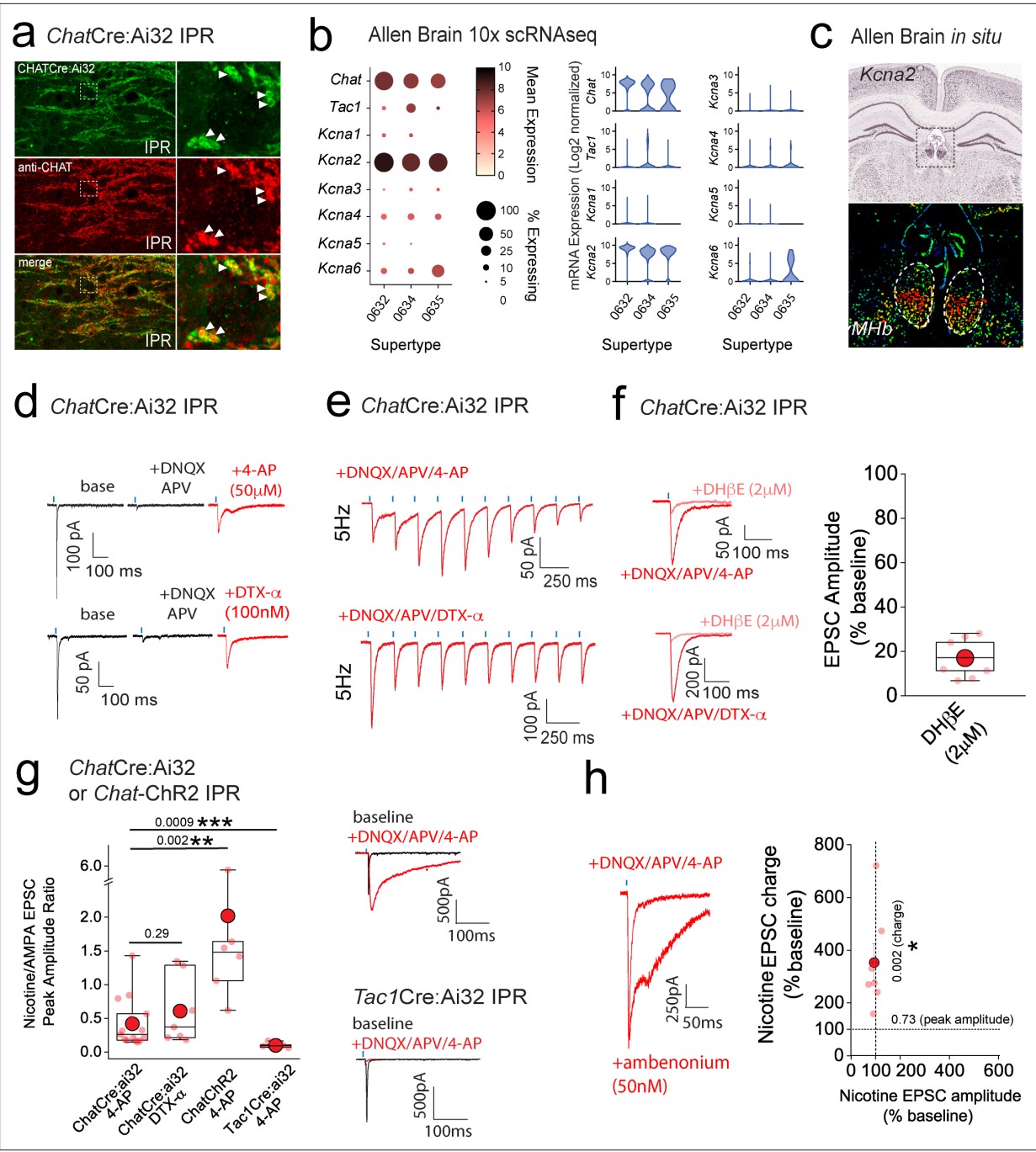

**Figure 7.** Kv1 channels constitute a molecular brake of nicotinic receptor-mediated signaling in the IPN. (**a**) High-resolution airy scan images of IPR in *Chat*Cre:Ai32 mice showing ChR2-expressing cholinergic boutons (green) and endogenous ChAT (red). Right panels are magnified regions of the boxed area in left panels. Arrows indicate faithful expression of ChAT within *Chat*Cre:Ai32 boutons (*Figure 5a*). (**b**) Corresponding dot and violin plots illustrating the relative expression of *Kcna-6* in *Chat* Supertypes only. (**c**) In situ hybridization for *Kcna2* mRNA (top panel) and corresponding pseudo-colored expression level (bottom panel) illustrating bias toward ventral medial habenula (mHb). Data are from the Allen Brain Institute (https://mouse.brain-map.org/gene/show/16263). (**d**) Voltage-clamp example traces of light-evoked EPSCs mediated by cholinergic mHb neurons (left panel) under baseline, following 10 μM DNQX/100 μM DL-APV plus 50 μM 4-AP (top panel) or plus 100 nM dendrotoxin-α (bottom panel). (**e**) Voltage-clamp example traces of 5 Hz trains of light-evoked EPSCs (10 stimuli) mediated by cholinergic mHb neurons in the presence of 10 μM DNQX/100 μM DL-APV and 100 μM 4-AP (top panel) or DTX-α (bottom panel) in a *Chat*Cre:Ai32 mouse. (**f**) Single light stimulus evoked EPSC in the presence of 10 μM DNQX/100 μM DL-APV and 100 μM 4-AP (top left panel) or DTX-α (bottom left panel) in the absence or presence of 2 μM DHβE. Box plot with individual data of the percentage inhibition of the light-driven EPSC peak amplitude mediated by cholinergic neurons in the presence of 10 μM DNQX/100 μM DL-APV/50 μM 4-AP or DTX-α (*n* = 9 recorded IPR neurons from 9 mice). (**g**) Box plot of the nicotine/AMPA (nAChR/AMPA) peak amplitude ratio within

*Figure 7 continued*

individual recorded IPR neurons percentage as measured under baseline (AMPA EPSC) and in the presence of 10 µM DNQX/50 µM DL-APV/100 µM 4-AP (nAChR EPSC; *n* = 15 recorded IPR neurons from 15 *Chat*Cre:Ai32 mice) or 100 nM DTX-α (nAChR EPSC; *n* = 7 recorded IPR neurons from 5 *Chat*Cre:Ai32 mice). nAChR/AMPA peak EPSC amplitude ratios were also performed in *Chat*-ChR2 (*n* = 6 recorded IPR neurons from 6 mice) and *Tac1*Cre:Ai32 mice (*n* = 6 recorded IPR neurons from 6 mice). Voltage-clamp example trace of light-evoked EPSCs under baseline and after addition of 10 µM DNQX/50 µM DL-APV/100 µM 4-AP in *Chat*-ChR2 (top right panel) and *Tac1*Cre:Ai32 (bottom right panel). (**h**) Voltage-clamp example trace of light-evoked nAChR-mediated EPSCs mediated by cholinergic mHb neurons under baseline and following application of 50 nM ambenonium (left panel) in a *Chat*ChR2 mouse. Scatter plot of the individual (light red symbols) and pooled (red symbol) percentage change in nAChR EPSC peak amplitude versus EPSC charge (measured over the first 500 ms duration of the EPSC) in each individual recording (*n* = 7 recorded IPR neurons in 5 mice; right panel). Data in (**b**) and (**c**) are from the publicly available Allen Brain Cell Atlas (https://knowledge.brain-map.org/abcatlas) and the Allen Brain Map (Allen Brain; https://mouse.brain-map.org/gene/show/16263), respectively. See methods for further details.

The online version of this article includes the following figure supplement(s) for figure 7:

**Figure supplement 1.** Effect of DNQX on synaptic transmission mediated by medial habenula (mHb) cholinergic neurons onto IPN prior to or after mu-opioid receptor (mOR) activation and during high frequency stimulation.

of any nAChR-mediated EPSCs in the presence of DNQX (*Figure 7—figure supplement 1d, e*). Finally, the potentiated ESPC following mOR activation is purely AMPAR mediated, as evidenced by complete block by DNQX (*Figure 7—figure supplement 1f–h*). Together, our functional analyses demonstrate that brief pulses of light (1–5 ms) in *Chat*Cre:Ai32 mice do not result in sufficient ACh release, if any, to elicit measurable postsynaptic nAChR responses in response to high frequency stimulus regimens and even under conditions where mOR activation strongly enhances glutamatergic transmission.

Interestingly, cholinergic transmission at the neuromuscular junction can be boosted via potassium channel blockade with fampridine (4-aminopyridine) and has been clinically indicated in disorders such as myasthenia gravis and multiple sclerosis (*Kostadinova and Danchev, 2019*). Furthermore, low micromolar 4-AP in slices amplifies acetylcholine release, as assessed by Grab-ACh-mediated fluorescence in the IPN (*Jing et al., 2018*; *Jing et al., 2020*). Experimentally, these concentrations are relatively selective in inhibiting $K^+$-channels of the Kv1 family. Probing the publicly available 10X single-cell RNAseq database provided by the Allen Brain Institute reveals that *Kcna2* (the gene encoding Kv1.2) is the most prevalently expressed in the designated *Chat* Supertypes (*Figure 1d* and *Figure 7b*; mean expression levels including zero values = 9.1, 6.8, and 4.1 in Supertypes 0632, 0634, and 0635, respectively). Furthermore, spatial interrogation via in situ hybridization (Allen Brain; https://mouse.brain-map.org/gene/show/16263) demonstrates the presence of appreciable *Kcna2* transcript levels in ventral habenula (*Figure 7c*) a region that is enriched in cholinergic neurons (*Figure 1b, c*). We therefore speculated that Kv1.2 block could result in enhancement of ACh release to perhaps reveal postsynaptic nAChR EPSCs in IPR neurons. Indeed, following the block of glutamatergic transmission (DNQX + APV), subsequent application of 4-AP (50–100 µM) or the more selective Kv1 channel antagonist, dendrotoxin-α (100 nM), results in the emergence of an EPSC elicited by brief single light pulses (*Figure 7d*) and in response to trains of 5 Hz stimulation (*Figure 7e*). This emergent EPSC is mediated by nAChRs as evidenced by ~80–90% block by the nicotinic receptor antagonist, DHβE (*Mulle et al., 1991*; *Morton et al., 2018*) (2 µM; *Figure 7f*).

Interestingly, this experimental approach allows for the calculation of nAChR;AMPAR peak amplitude ratio enabling an assessment of fast excitatory cholinergic transmission normalized to stimulus intensity required to produce a given AMPA EPSC response in an individual postsynaptic IPR neuron. Thus, the size of the nAChR response can be directly compared across multiple recordings from differing slices, mice, and experimental conditions. We first assessed the extent to which both 4-AP and DTX-α reveal light-evoked nAChR EPSCs, finding no significant difference in the nAChR:AMPAR ratio, demonstrating that the specific Kv1.1/Kv1.2 channel antagonist unmasks nAChR-mediated synaptic transmission in a similar manner to low concentration 4-AP (*Figure 7g*). This, in combination with the predominant expression of *Kcna2* (*Figure 7b and c*) and relative lack of other *Kcna* transcripts (*Figure 7b*) in mHb ChAT neurons, confirms our hypothesis that Kv1.2 plays a central role in unmasking nAChR-mediated synaptic transmission. In addition to *Chat*Cre:Ai32 mice, we also employed the *Chat*-ChR2 mouse. Not surprisingly, nAChR EPSCs could also be elicited in this mouse line following 4-AP application (*Figure 7g*). However, the measured nAChR:AMPAR ratio is significantly higher than that seen in *Chat*Cre:Ai32 mice (*Figure 7g*), indicating a skew toward more

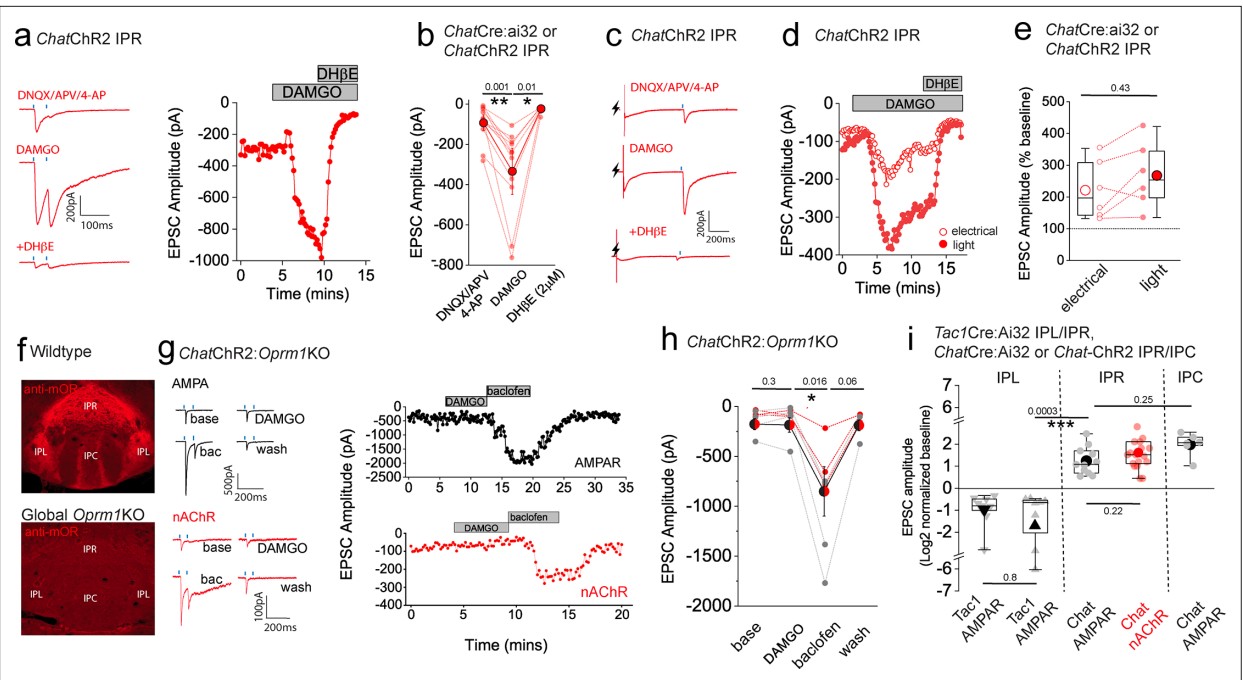

**Figure 8.** Mu-opioid receptor (mOR) potentiates nAChR EPSC amplitude revealing an interplay between opioid and cholinergic systems in the habenulo-interpeduncular axis. (**a**) Voltage-clamp example traces of light-evoked nAChR EPSCs (two stimuli at 20 Hz; blue dashes) mediated by cholinergic medial habenula (mHb) neurons (left panel) and time course of peak amplitude (right panel) under baseline, 500 nM DAMGO, and 500 nM DAMGO plus 2 μm DHβE conditions. (**b**) Individual and mean (red filled symbols) data of nAChR EPSC peak amplitude. (**c, d**) Voltage-clamp example traces of simultaneous electrical (lightning symbol) and light-evoked (blue dash) nAChR EPSCs and time course of peak amplitude (open and filled symbols representing electrical and light-evoked peak amplitude of nAChR EPSCs, respectively) under baseline, 500 nM DAMGO, and 500 nM DAMGO plus 2 μm DHβE conditions. (**e**) Percentage change in electrical and light-evoked nAChR EPSC peak amplitude elicited by 500 nM DAMGO in each individual recorded IPR neuron (*n* = 6 recorded neurons from 6 mice). (**f**) Confocal images of endogenous mOR protein expression in IPN of WT and homozygote *Oprm1* KO mice. (**g**) Voltage-clamp example traces of light-evoked AMPA and nAChR EPSCs (two stimuli at 20 Hz; blue dashes) mediated by cholinergic mHb neurons (top and bottom left panels, respectively) and time course of peak amplitude (right panels) under baseline, 500 nM DAMGO, 1 μM baclofen, and washout conditions in *Chat*ChR2: *Oprm1*KO mice. (**h**) Individual and mean data of light-evoked AMPAR (black symbols; *n* = 4 from 4 mice) and nAChR (red symbols; *n* = 3 from 3 mice) peak amplitude in response to DAMGO and baclofen application in *Chat*ChR2: *Oprm1*KO mice. Note in two of the four AMPAR EPSC recordings, washout of the baclofen effect was not performed. (**i**) Summary box plot with individual data of the percentage normalized changes (log₂) of AMPAR and nAChR peak amplitudes in response to mOR activation mediated by SP and cholinergic neurons in various subdivisions of IPN tested. Note that the normalized data in this panel are replotted from the absolute peak EPSC amplitude changes mediated by mOR agonists in **Figure 2b**, **Figure 3b, f**, **Figure 5e** and **Figure 7b** and **Figure 3—figure supplement 1b**.

prominent cholinergic transmission when compared to that mediated by glutamate. It is unclear as to the exact reason for this divergence, but it may be related to increased levels of the vesicular Ach transporter (vAChT) expression observed in *Chat*:ChR2 mice (**Kolisnyk et al., 2013**). Although *Tac1* mHb neurons also express *Kcna2* transcript, albeit lower than in the ChAT neurons (data not shown), 4-AP application following block of glutamatergic transmission does not result in emergence of any additional EPSC in response to single stimuli in *Tac1*Cre:Ai32 mice (**Figure 7g**). Finally, application of a cholinesterase inhibitor (ambenonium, 50 nM) markedly prolongs the EPSC waveform with minimal effect on amplitude (**Figure 7h**). Thus, together, these results corroborate the role of Kv1.2 as a molecular brake, removal of which results in the emergence of habenulo-interpeduncular synaptic transmission via postsynaptic nAChR signaling on IPR GABAergic neurons.

Having established the ability to reliably assess an otherwise reluctant cholinergic transmission, we next investigated how nAChR-mediated signaling in the IPR is impacted by mOR activation. As with AMPAR-mediated EPSCs, DAMGO produces a marked potentiation of light-evoked nAChR EPSC amplitude in IPR, with the augmented EPSC being completely blocked by DHβE (**Figure 8a, b**). The mOR-mediated potentiation of both the glutamatergic and cholinergic output is essentially similar (**Figure 8i**). Till now, we have utilized optogenetic approaches to assay synaptic transmission in the IPN. Interestingly, following acute block of Kv1, nAChR EPSCs could also be elicited via electrical

stimulation (*Figure 8c*). Further, DAMGO reliably potentiated both electrical and light-evoked nAChR EPSCs in a single IPR neuron to similar extents, a response that is effectively blocked by DHβE (*Figure 8c–e*). Finally, we tested the molecular specificity of the DAMGO response employing *Chat-ChR2*:global *Oprm1* knockout mice. In these mice, DAMGO application is ineffective in augmenting glutamatergic or cholinergic-mediated neurotransmission in IPR, while the potentiation upon GABA$_B$ receptor activation previously described (*Koppensteiner et al., 2024*; *Bhandari et al., 2021*; *Melani et al., 2019*) remains intact (*Figure 8f–h*; note that in these experiments, CGP55845A, which is routinely included in all other experiments, was omitted). This genetic approach complements our previous pharmacological data (*Figure 3g*) to directly implicate mORs in the potentiation of mHb-IPN transmission. The overall effects of mOR activation on synaptic transmission mediated by the distinct mHb afferent systems in the various subdivisions of IPN tested are summarized in *Figure 8i*.

## Discussion

In the current study, we reveal a remarkable augmentation of glutamatergic/cholinergic co-transmission by mHb cholinergic neurons following mOR activation in the IPN. However, the underlying cellular/network mechanisms responsible remain unclear. The most parsimonious explanation is that mORs in proximity to the release machinery of cholinergic nerve terminals (*Gardon et al., 2014*) mediate this effect. However, one must consolidate the fact that a Gi-linked receptor, which typically serves to directly inhibit release, results in a seemingly paradoxical potentiation. Functional and mechanistic studies have elegantly demonstrated that GABA$_B$ receptors (another member of the Gi-linked subfamily) trigger acute molecular and structural adaptations resulting in enhanced Ca$^{2+}$ influx and a switch of transmission modes within single presynaptic terminals to ultimately increase neurotransmitter release (*Koppensteiner et al., 2024*; *Zhang et al., 2016*). Although these observations set a precedent for potentiation by Gi-linked receptor activation in the IPN, similar experimental approaches to that employed in the examination of the GABA$_B$-receptor-mediated potentiation (*Koppensteiner et al., 2024*; *Zhang et al., 2016*; *Jing et al., 2020*) are required to definitively implicate overlapping mechanisms. In other brain regions, a canonical inhibitory influence of mORs elicits overall network excitation via disinhibition (*Chen et al., 2015*; *McQuiston and Saggau, 2003*; *Johnson and North, 1992*; *Lau et al., 2020*). Furthermore, numerous studies have revealed an intricate interplay involving diverse neuromodulatory components intrinsic to the IPN that serve to regulate synaptic transmission (*Ables et al., 2017*; *Melani et al., 2019*; *Koppensteiner et al., 2017*). Thus, taking the results of the current study in isolation, we cannot discount the possibility that the potentiation observed here may result via the activation of mORs residing on other neural elements within the IPN microcircuitry such as postsynaptic neurons or glia, for instance. However, it must be noted that a recent study employing a viral strategy to elicit conditional knockout of mORs in *Oprm1*-expressing mHb neurons specifically prevents the DAMGO-mediated potentiation of glutamatergic transmission onto the IPN, indicating a presynaptic locus (*Singhal et al., 2025*). Regardless of the exact underlying cellular and/or network mechanisms, our data extend the previously described Gi-linked receptor potentiation of neurotransmission in the IPN (*Koppensteiner et al., 2024*; *Zhang et al., 2016*; *Jing et al., 2020*) to mORs, a predominant target of a societally relevant and prevalently misused class of drugs. It would be of interest to determine whether GABA$_B$ receptor activation exerts a similar inhibitory influence as mORs on the newly discovered SP neuronal-mediated transmission to the IPR. In addition, does the modulation of ChAT neuronal glutamatergic output by GABA$_B$ receptors undergo a similar developmental regulation to that observed with mORs? These additional functional comparisons of the synaptic influences of these two distinct Gi-linked receptors may shed light as to the similarity, or lack thereof, regarding the respective underlying cellular mechanisms.

Recent generation of transgenic mice (*Bailly et al., 2020*; *Weibel et al., 2013*; *Mengaziol et al., 2022*) has permitted manipulation of mOR-expressing neurons and mOR receptors including those expressed specifically within the mHb/IPN. Emerging behavioral studies adopting such conditional genetic approaches have highlighted an important role of mORs within this circuitry in mediating varying aspects of reward and aversion (*Allain et al., 2022*; *Boulos et al., 2020*). Unsurprisingly, it has been concluded that the observed behavioral effects of such manipulations are due to perturbations of a generalized inhibition mediated by mORs within the mHb/IPN circuit. Here, we reveal a functional dichotomy characterized by an inhibitory yet excitatory influence on neurotransmission mediated by SP and cholinergic neurons, respectively. These distinct neuronal populations participate in and

drive specific behavioral facets demonstrating a division of labor. For example, SP neuronal activity positively correlates with reward outcome, history, and hedonic value (*Sylwestrak et al., 2022*; *Hsu et al., 2014*), whereas that of cholinergic neurons is reduced during behaviors associated with reward (*Sylwestrak et al., 2022*). mHb cholinergic neurons have also been linked to negative affect including anxiety and depression (*Cho et al., 2020*; *Seigneur et al., 2018*; *Pang et al., 2016*) that promote drug-seeking behaviors during withdrawal (*Batalla et al., 2017*). We reveal for the first time that the IPR subdivision receives functional glutamatergic afferent input from SP neurons that, based on previous anatomical interrogation, were considered to exclusively target the IPL (*Quina et al., 2017*). Interestingly, the novel opposing effect on transmission extends to the IPR, thus demonstrating a role for mORs in altering the relative salience of these distinct afferent systems with regard to the recruitment of common downstream IPR neurons. The IPR houses SST GABAergic neurons whose afferents impinge on the raphe nucleus and lateral dorsal tegmental nucleus, the latter influencing the VTA and, in turn, its downstream structures (*Ables et al., 2017*; *Hsu et al., 2013*; *Monical and McGehee, 2025*). Thus, careful consideration regarding future examination of behavioral outcomes resulting from the described contrasting effects of mOR activation on the parallel processing of reward (mHb SP) and anti-reward (mHb ChAT) in this circuit is essential.

A limitation of the current study arises from the sole utilization of a transgenic approach to selectively assay light-evoked synaptic transmission from ChAT and SP neuronal populations, respectively. Although the mHb provides most of the afferent input to the IPN, this approach does not exclude possible activation of additional inputs from other brain regions (*Liang et al., 2024*; *Lima et al., 2017*; *Bueno et al., 2019*). This caveat can be circumvented by use of stereotaxic viral delivery to express ChR2 solely in the mHb (*Souter et al., 2022*; *Singhal et al., 2025*; *Hsu et al., 2013*). However, it is unclear if neuronal inputs from these possible alternate sources (*Liang et al., 2024*; *Lima et al., 2017*; *Bueno et al., 2019*) are glutamatergic in nature and mediated by a *Tac1/Oprm1*-expressing neuronal population. Nevertheless, to definitively identify the described novel input as one that originates from mHb SP neurons will require the future use of such viral strategies.

Throughout this study, we contextualize the effect of mOR activation in the IPN primarily through the lens of exogenously introduced opioids. However, physiological activation of mORs can be mediated to differing extents by endogenous ligands such as dynorphin or enkephalin. Indeed, we demonstrate that met-enkephalin effectively boosts co-transmission from cholinergic neurons in a similar manner to morphine. Interestingly, the IPN is home to a population of pro-enkephalin (PENK)-expressing neurons and detectable met-enkephalin immunoreactivity (*Gardon et al., 2014*; *Mahalik and Finger, 1986*). In other brain regions, manipulation of PENK neuronal activity and/or ablation of PENK itself elicits mOR-mediated circuit modulation and behavioral alterations (*You et al., 2023*; *Castro et al., 2021*; *Leroy et al., 2022*). It is anticipated that activation of mORs by local endogenous opioid release yields similar complex effects to those following exogenous application of mOR agonists described here. Thus, this physiological route will have major implications concerning the role of mHB/IPN mORs in mediating the positive reinforcing effects of non-opioid drugs of abuse (e.g. nicotine, alcohol, amphetamines) and other 'natural' rewarding stimuli (e.g. such as those associated with social interaction, exercise, and food intake), thus extending the relevance of our study to these additional modalities.

Another novel finding in the current study relates to the identification of a 'molecular brake' that exerts strong control over nAChR signaling impinging on postsynaptic IPN neurons. Specifically, compromising the function of delayed rectifying $K^+$-channels containing Kv1.2 subunits results in the emergence of nAChR EPSCs in response to light and electrical stimuli delivered with physiological paradigms likely through increased axonal/bouton excitability and calcium influx via action potential waveform modulation. Kv1.2 function can be bidirectionally impacted by use-dependent mechanisms or through secondary activity-mediated cascades resulting in post-translational modifications (e.g. phosphorylation state) that impact its function and/or trafficking (*Baronas et al., 2015*; *Thayer et al., 2016*; *Yang et al., 2007*). These plausible endogenous mechanistic routes could serve to titer the strength nAChR-mediated transmission in the IPR.

The resident nAChRs in the mHb/IPN are primarily encoded by the *CHRNA3/B4/A5* gene cluster and the role of this circuitry in nicotine use is well characterized. For example, increased propensity for nicotine abuse in humans is associated with dysregulation of nAChR function precipitated by single nucleotide polymorphisms of this cluster (*Amos et al., 2008*; *Improgo et al., 2010*; *Brynildsen and*

*Blendy, 2021*). Experimental manipulation of nAChR signaling in the mHb/IPN generates phenotypes associated with various aspects of nicotine consumption in mouse models (*Morton et al., 2018*; *Antolin-Fontes et al., 2015*; *Frahm et al., 2015*; *Fowler et al., 2011*; *Mathis and Kenny, 2019*). Furthermore, chronic nicotine results in adaptations in nAChR expression and function that can exacerbate continued nicotine consumption (*Arvin et al., 2019*; *Mondoloni et al., 2023*; *Govind et al., 2009*; *Jin et al., 2020*). Together, these studies establish a link between nAChR-mediated signaling within this circuitry and prevalence of nicotine misuse. Here we highlight an intriguing interplay between opioid and cholinergic systems within the mHb/IPN axes. Our data clearly demonstrate that mORs and Kv1.2 together comprise synergistic molecular targets that, in addition to others previously identified (*Bhandari et al., 2021*), when leveraged in tandem can manipulate nAChR-mediated signaling to yield potential interventions for nicotine overuse.

Adolescence is a critical period for many aspects of brain and social development and represents a high-risk demographic group for drug use developing into long-term addiction (*Ahmadi-Soleimani et al., 2024*). Propelled by the societal introduction of highly potent synthetic morphine analogs (e.g. fentanyl), one of the many devastating sequelae of the well-documented opioid crisis is an alarming high rate of overdose deaths not only in adults but also during vulnerable teenage years. Of note is the striking valence conversion from depression to potentiation of mHb ChAT neuronal signaling in the IPR occurring around the late postnatal stages. This is in stark contrast to that seen with the newly described SP neuronal-mediated transmission in this same IPN subdivision where mOR activation is inhibitory at all life stages assayed. Thus, during postnatal development, opioids acting via mORs result in a blanket inhibition of these two distinct inputs prior to the establishment of the afferent specific modulation in the adult. Interestingly, numerous studies have highlighted changes in the role of mORs in reward-associated behaviors coinciding with similar periods (*Harda et al., 2023*; *King'uyu et al., 2024*). An attractive hypothesis is that coordinated modifications at the cellular, molecular, and/or network level underlie this dynamic developmental regulation. Thus, our circuit analyses provide insights into the neural correlates for the differing propensity of mORs to regulate not only positive reinforcement but also the negative effects associated with withdrawal at various stages of development (*Wang, 2019*). Whether these adaptations underlie the increased propensity for substance abuse during development remains to be ascertained. Furthermore, opioid exposure during early developmental epochs imparts long-lasting effects on the brain's reward circuitry (*Salmanzadeh et al., 2020*). Thus, together, it is imperative to consider the newly described developmentally regulated impact of mORs in the modulation of the mHb/IPN circuitry. This is of particular relevance with respect to the generation of potential preventative treatments for the deleterious consequences of fetal opioid exposure and SUDs in high-risk juveniles.

In addition to a central role in addiction, the mHb/IPN circuitry encodes fear-associated behaviors (*Koppensteiner et al., 2024*; *Zhang et al., 2016*; *Fernández-Suárez et al., 2021*; *Soria-Gómez et al., 2015*; *Roy and Parhar, 2022*). Interestingly, prevention of GABA$_B$-receptor function in mHb cholinergic neurons or increasing IPN neuronal activity facilitates and augments expression of fear extinction, respectively (*Koppensteiner et al., 2024*; *Zhang et al., 2016*). Dysregulation of this behavioral aspect is a key underlying cause of post-traumatic stress disorder (PTSD) (*Careaga et al., 2016*). Based on this previous work, the novel mOR-mediated potentiation of the mHb cholinergic transmission and hence IPN neuronal recruitment described here constitutes a parallel, yet alternative circuit mechanism involved in the extinction of learned fear. Thus, pharmacogenetic manipulation of mOR-mediated signaling specifically in the IPN may constitute a viable approach to alleviate conditions associated with abnormal fear processing such as those seen in PTSD.

In summary, we highlight several previously undescribed and hence unappreciated roles of mORs in the dynamic regulation of synaptic transmission impinging on IPN GABAergic neurons following exposure to exogenous opioids. Although additional studies are needed to reveal the underlying cellular mechanisms responsible for these complex, divergent facets of opioid-mediated effects in the habenulo-interpeduncular axis, the current detailed functional analyses lay out a valuable roadmap. One that necessitates consideration in future interrogation concerning the role of these receptors in this relatively understudied brain circuitry that encodes aspects of emotion, hedonic, and addiction behaviors.

## Footnote

While conducting our current study, we became aware of one from the Hnasko laboratory also investigating the role of opioids on habenulo-interpeduncular synaptic transmission (*Singhal et al., 2025*). Their observations regarding a similar mOR-mediated potentiation of glutamatergic signaling by mHb cholinergic neurons point to the reproducibility of this unexpected, novel, and potentially important circuit mechanism.

# Materials and methods

## Animals

All procedures in the current study involving experimental mice were conducted in strict accordance with an active animal protocol (ASP#23-045) approved by the Animal Care Use Committee of the National Institute of Child Health and Human Development. All transgenic mice were attained from The Jackson Laboratory (ME, USA) and were as follows: *Chat*Cre (strain #031661), *Tac1*Cre (strain #021877), SSTCre (strain #018973); Ai9 (strain #007909), Ai32 (strain #024109), *Chat*-ChR2-EYFP (strain #014546); *Oprm1* KO (strain #007599). For conditional expression of td-Tomato (Ai9) or ChR2 (Ai32), both male and female breeders were homozygous, and all immunocytochemical or electrophysiological experiments were performed in their progeny that were heterozygous for both the Cre and floxed alleles. Heterozygous *Chat*-ChR2 mice were used throughout the study either alone or crossed with the *Oprm1* KO mouse (homozygous) or SSTCre:Ai9 (heterozygous for both alleles). Both male and female mice were investigated, and the data were pooled.

## Reagents

Pharmacological reagents used in this study are as follows: DNQX (Cat. No. 2312/10), DL-APV (Cat. No. 0105/10), CGP55845A (Cat. No. 1248/10), Picrotoxin (Cat. No. 1128), bicuculline (Cat. No. 0109/10)**,** DAMGO (Cat. No. 1171/1), CTAP (Cat. No. 1560/1), and DHβE (Cat. No. 2349) were all purchased from Biotechne/Tocris (MN, USA); Dendrotoxin-α (Cat No. D-350) was purchased from Alomone Labs (Jerusalem, Israel). Met-enkephalin (Cat. No. M6638) was purchased from MilliporeSigma (MA, USA). Ambenonium (Cat. No. sc-203507) was purchased from Santa Cruz Biotechnology (TX, USA). Morphine Sulfate (Cat. No. NDC: 0641-6125) was from Hikma (NJ, USA) and attained via the NIH Division of Veterinary Resources (DVR) Pharmacy.

Primary and corresponding secondary antibodies used in this study for immunocytochemistry and the working dilutions employed are as follows: Guinea pig anti-RFP (Synaptic Systems, Cat. No. 390005, 1:500) with CF555-conjugated Donkey anti-Guinea Pig IgG (Biotium Cat. No. 20276, 1:1000). Chicken anti-GFP (AvesLabs, Cat. No. GFP-1010, 1:1000) with CF488-conjugated Donkey anti-Chicken IgY (Biotium Cat. No. 20166, 1:1000 working dilution). Rabbit anti-MOR (ABCAM, Cat. No. ab134054, 1:500) with CF555-conjugated Donkey anti-Rabbit IgG (Biotium cat#20038, 1:1000). Goat anti-ChAT, Millipore (Cat. No. AB144P, 1:1000) with CF555-conjugated Donkey anti-Goat IgG (Biotium, Cat. No. 20039, 1:1000 working dilution).

## Immunocytochemistry and imaging

Mice (P20 and older) were perfused trans-cardinally using 4% PFA and dissected brain tissues were post-fixed in 4% PFA for 24 hr at 4°C. For P10 mice, brains were removed and drop-fixed in 4% PFA for 24 hr at 4°C. P10 mouse brains were not perfused but brains were removed and drop-fixed as stated above. Fixed brain tissues were thoroughly washed in 1x phosphate buffer (PB) followed by cryopreservation using 30% sucrose. 50 μm coronal sections were made on a frozen microtome. To perform floating section IHC, brain slices were washed with 1x PB at room temperature for 1 hr with two to three changes of 1x PB followed by blocking and permeabilizing in Blocking Solution (1x PB +10% donkey serum + 0.5% Triton X-100) at room temperature for at least 2 hr. Blocked brain slices were incubated in primary antibodies, which were diluted to working concentration using Antibody Solution (1x PB + 1% donkey serum + 0.1% Triton X-100), at 4°C for 24–48 hr. After washing with 1x PB at room temperature for 15 min with three repeats, brain slices were incubated in secondary antibodies diluted with Antibody Solution at room temperature for 1 hr, followed by DAPI staining at 1 μg/ml for additional 15 min. After being washed with 1x PB at room temperature for 15 min with three repeats, brain slices were mounted on gelatin-coated slides followed by air drying, covered with

No. 1.5 cover slides ProLong Glass Antifade Mountant (Therm Fisher Scientific, Cat#P36984), cured in darkness overnight before imaging. Confocal images were acquired on a Zeiss LSM900 using ZenBlue software. 10x and 20x confocal images of whole habenula and IPN were acquired as tiles with z-steps of 3 µm × 5 and 0.9 µm × 20, respectively. Airy scan images were acquired at 63x using multiplex 2Y mode to achieve super resolution. 2 µm depth of images were captured at z-steps of 0.2 µm × 10, with or without tiling. All comparable images of tissues with different genotypes or developmental stages were acquired using the same light path configuration, channel input, gain, and offset. Images were imported into Fiji and Adobe Photoshop for processing and densitometry measurements where applicable.

## Electrophysiology

P15–P91 mice (for details refer to the results section) were anesthetized with isoflurane, and the brain removed in ice-cold partial sucrose substituted ASCF (ssACSF) solution containing (in mM): 90 sucrose, 80 NaCl, 25 $NaHCO_3$, 1.25 $NaH_2PO_4$, 3.5 KCl, 4.5 $MgCl_2$, 0.5 $CaCl_2$, 10 glucose, saturated with 95% $O_2$ and 5% $CO_2$ (pH 7.4; osmolality 300–310 mOsm). Coronal sections containing either the mHb or IPN (270–300 µm) were cut using a VT-1000S vibratome (Leica Microsystems, Bannockburn, IL) in ice-cold ssACSF. The slices were allowed to recover in the ssACSF at 31–33°C for 20–30 min followed by an additional 20 min at room temperature. Slices were then transferred to our standard extracellular solution (ACSF) of the following composition (in mM): 130 NaCl, 24 $NaHCO_3$, 3.5 KCl, 1.25 $NaH_2PO_4$, 2.5 $CaCl_2$, 1.5 $MgCl_2$, and 10 glucose, saturated with 95% $O_2$ and 5% $CO_2$ (pH 7.4; osmolality 300–310 mOsm) for storage until electrophysiological recording. Slices were transferred to an upright Olympus BX51WI microscope and visualized with infrared differential interference contrast microscopy and perfused (2–4 ml/min) with the above extracellular solution. Unless otherwise indicated, the recording extracellular solution was routinely supplemented with 2 µM CGP55845A hydrochloride (Biotechne/Tocris Cat#1248), 50 µM picrotoxin (Biotechne/Tocris Cat#1128), and 10 µM bicuculline methobromide (Biotechne/Tocris Cat#0131).

All recordings were performed at 31–33°C with electrodes (3–5 MΩ) pulled from borosilicate glass (World Precision Instruments, Sarasota, FL) filled with one of two intracellular solutions (in mM): (1) 135 CsMeSO$_4$, 8 KCl, 4 MgATP, 0.3 NaGTP, 5 QX-314, 0.5 EGTA, and 3 mg/ml biocytin. (2) 150 K-gluconate, 3 $MgCl_2$, 0.5 EGTA, 2 MgATP, 0.3 $Na_2$GTP, 10 HEPES, and 3 mg/ml biocytin. pH was adjusted to 7.3 with KOH or CsOH and osmolality adjusted to 285–295 mOsm. Whole-cell patch-clamp recordings were made using a Multiclamp 700B amplifier (Molecular Devices, Sunnyvale, CA). Signals were filtered at 4–10 kHz and digitized at 10–20 kHz (Digidata 1322A and pClamp 10 Software; Molecular Devices, Sunnyvale, CA). For cell-attached recordings, pipettes were filled with our standard extracellular solution.

For optogenetic stimulation of ChR2 light stimuli at a wavelength of 470 nm and duration of 1–5 ms were delivered through the 40X water immersion objective using a CoolLED pE-4000 Illumination system (Andover, UK). Arbitrary LED power typically ranged between 10% and 100% corresponding to ~0.4–6.9 mW/mm$^2$ as measured with a digital power meter (Thor Labs, NJ, USA). Where applicable, electrical stimulation (0.1ms; typically, 100–700 µA) was performed by placement of a tungsten bipolar electrode in the IPR (World Precision Instruments, FL, USA; Catalog # TST33A05KT) and a constant current stimulus isolator (World Precision Instruments; Catalog # A385 or A365). sEPSC analyses were performed on post hoc filtered episodic traces (Clampfit; Bessel 8-pole, –3 db cutoff 1000 Hz). Frequency of sEPSCs and average sEPSC amplitude were determined over five consecutive sweeps totaling a period of 4 s (i.e. 0.8 s per sweep). sEPSCs were detected via a thresholding procedure set at four times the RMS baseline noise. Note that only neurons with a baseline sEPSC frequency of >8 Hz were selected for further interrogation. Analyses of electrophysiological data were performed in Clampfit (Molecular Devices, Sunnyvale, CA) and curated in Microsoft Excel spreadsheets.

## Analyses of publicly available single-cell RNA sequencing transcriptomic data

We utilized the Allen Institute dataset of the single-cell transcriptomes of ~4 million cells across the entire mouse brain (*Yao et al., 2023*). To probe the region and genes of interest, we adapted Python notebooks provided for accessing the Allen Brain Cell (ABC) Atlas (https://alleninstitute.github.io/abc_atlas_access/intro.html). The log$_2$-normalized expression matrices were downloaded and filtered

to reach the cell types of interest within the habenula class '17 MH-LH Glut' (~10.8 k cells). For the current study, we focused our analyses on the mHb Subclass '145 MH Tac2 Glut' (~8k cells). Within this subclass, we probed the five supertypes '0632–0636 MH Tac2 Glut', querying the genes of interest: *Chat*, *Tac1*, *Slc17a7*, *Kcna1-6*, and *Oprm1*. *Chat* supertypes were considered those with mean *Chat* expression including zero values >3 (0632, 0634, and 0635; 6803 cells), while *Tac1* supertypes were defined as those with mean *Tac1* expression including zero values >3 (0633; 878 cells). Violin and dot plots depicting gene expression levels were created in GraphPad Prism.

## Statistical analyses

Tests for normality (Shapiro–Wilk test) were performed on all datasets to be compared prior to statistical analyses and appropriate parametric (unpaired or paired *t*-tests) or non-parametric (Mann–Whitney *U* or Wilcoxon signed rank tests) were accordingly performed. Exact p-values are stated within the figures and values of $p < 0.05$, $p < 0.01$, or $p < 0.001$ were deemed statistically significant and designated with 1, 2, or 3 asterisks, respectively. Error bars in the figures represent standard error of the means unless otherwise indicated. Box whisker plots were constructed as follows: symbol denotes mean value; line represents the median value; lower and upper box limits represent 25th and 75th percentiles, respectively. Lower and upper whiskers represent minimal and maximal data points, respectively.

## Acknowledgements

We would like to thank Dr. Cole Malloy for critical discussion throughout the conduction of the study. This work was supported by an NICHD Intramural Research Program (IRP) grant to CJM (1ZIAHD001205-32). This research was supported by the Intramural Research Program of the National Institutes of Health (NIH). The contributions of the NIH author(s) are considered Works of the United States Government. The findings and conclusions presented in this paper are those of the author(s) and do not necessarily reflect the views of the NIH or the U.S. Department of Health and Human Services.

## Additional information

### Funding

| Funder | Grant reference number | Author |
|---|---|---|
| Eunice Kennedy Shriver National Institute of Child Health and Human Development | 1ZIAHD001205-32 | Chris J McBain |

The funders had no role in study design, data collection, and interpretation, or the decision to submit the work for publication.

### Author contributions

Ramesh Chittajallu, Conceptualization, Data curation, Formal analysis, Supervision, Investigation, Methodology, Writing - original draft, Project administration, Writing - review and editing; Anna Vlachos, Xiaoqing Yuan, Edra London, Investigation; Adam P Caccavano, Data curation, Formal analysis, Investigation, Writing - review and editing; Steven Hunt, Resources, Investigation; Daniel Abebe, Resources; Kenneth A Pelkey, Conceptualization, Supervision; Chris J McBain, Conceptualization, Supervision, Funding acquisition

### Author ORCIDs

Ramesh Chittajallu ⓘ https://orcid.org/0000-0002-9794-0052
Anna Vlachos ⓘ https://orcid.org/0000-0002-4411-9447
Adam P Caccavano ⓘ https://orcid.org/0000-0002-6819-533X
Kenneth A Pelkey ⓘ https://orcid.org/0000-0002-9731-1336
Chris J McBain ⓘ https://orcid.org/0000-0002-5909-0157

## Ethics

All procedures involving experimental mice were conducted in strict accordance with an active animal protocol (ASP#23-045) approved by the Animal Care Use Committee of the National Institute of Child Health and Human Development. Please refer to the methods sections for details of protocols employed.

Reviewer #1 (Public review): https://doi.org/10.7554/eLife.106062.3.sa1
Reviewer #2 (Public review): https://doi.org/10.7554/eLife.106062.3.sa2
Reviewer #3 (Public review): https://doi.org/10.7554/eLife.106062.3.sa3
Author response https://doi.org/10.7554/eLife.106062.3.sa4

# Additional files

### Supplementary files
MDAR checklist

### Data availability

All analyses of data associated with the manuscript have been made available: https://data.mendeley.com/datasets/3dtkmss9xz/2. Adapted Python notebooks employed for analyses and graphical representation of publicly available Allen Brain Cell Atlas mouse whole brain scRNA sequencing data can be accessed via the following link: https://github.com/acaccavano/ABC-Atlas_Chittajallu2025/ (copy archived at *Caccavano, 2025*).

The following dataset was generated:

| Author(s) | Year | Dataset title | Dataset URL | Database and Identifier |
|---|---|---|---|---|
| Chittajallu R | 2025 | Complex opioid driven modulation of glutamatergic and cholinergic transmission in a GABAergic brain nucleus associated with emotion, reward and addiction | https://doi.org/10.17632/3dtkmss9xz.2 | Mendeley Data, 10.17632/3dtkmss9xz.2 |

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
