## [Editor Report · eLife Assessment]

This study presents **important** information about the role of mu opioid receptors in neurotransmission between the medial habenula and the interpeduncular nucleus. The authors provide **convincing** evidence that mu opioid receptor activation has differential effects on transmission from substance P neurons and cholinergic neurons, and that blockade of potassium channels can unmask a nicotinic cholinergic synaptic response. This work will be of high interest to those studying this brain region, and potentially to the larger neuroscience community studying motivated behavior.

---

## [Referee Report · Reviewer #1 (Public review)]

Summary:

In this manuscript, the authors demonstrate for the first time that opioid signaling has opposing effects on the same target neuron depending on the source of the input. Further, the authors provide evidence to support the role of potassium channels in regulating a brake on glutamatergic and cholinergic signaling, with the latter finding being developmentally regulated and responsive to opioid treatment. This evidence solves a conundrum regarding cholinergic signaling in the interpeduncular nucleus that evaded elucidation for many years.

Strengths:

This manuscript provides 3 novel and important findings that significantly advance our understanding of the medial habenula-interpeduncular circuitry:

(1) Mu opioid receptor activation (mOR) reduces postsynaptic glutamatergic currents elicited from substance P neurons while simultaneously enhancing postsynaptic glutamatergic currents from cholinergic neurons, with the latter being developmentally regulated.

(2) Substance P neurons from the Mhb provide functional input to the rostral nucleus of the IPN, in addition to the previously characterized lateral nuclei.

(3) Potassium channels (Kv1.2) provide a break on neurotransmission in the IPN,

The findings here suggest that the authors have identified a novel mechanism for the normal function of neurotransmission in the IPN, so it would be expected to be observable in almost any animal. In the revised manuscript, the authors put forth significant effort to increase the n, thus increasing the confidence in the observations.

There are also significant sex differences in nAChR expression in the IPN that might not be functionally apparent using the low n presented here. In the revised manuscript, the authors increased the n, and provided data to the reviewers that no significant sex differences were apparent, although there was a trend. Future studies should examine sex differences in detail.

There are also some particularly novel observations that are presented but not followed up on, and this creates a somewhat disjointed story. For example, in Figure 2, the authors identify neurons in which no response is elicited by light stimulation of ChAT-neurons, but application of DAMGO (mOR agonist) un-silences these neurons. Are there baseline differences in the electrophysiological or morphological properties of these "silent" neurons compared to the responsive neurons? In the revised manuscript, the authors directly tested this with new experiments in SST+ neurons in the IPN, demonstrating convincingly that mOR activation unsilences these neurons.

With the revisions, the authors have addressed the reviewers' concerns and significantly improved the manuscript. I find no further weaknesses.

---

## [Referee Report · Reviewer #2 (Public review)]

Summary:

In this paper, Chittajallu and colleagues present compelling evidence that mu opioid receptor (MOR) activation can potentiate synaptic neurotransmission in a medial habenula to interpeduncular nucleus (mHb-IPN) subcircuit. While, projections from mHb tachykinin 1 (Tac1) neurons onto lateral IPN neurons show a canonical opioid-induced synaptic depression in glutamate release, excitatory neurotransmission in mHb choline acetyltransferase (ChAT) projections to the rostral IPN is potentiated by opioids. This function emerges around age P27 in mice, when MOR expression in the IPN peaks.

Strengths:

Carefully executed electrophysiological experiments with appropriate controls. Interesting description of a neurodevelopmental change in the effects of opioids on mHb-IPN signaling.

Weaknesses:

A minor concern is that the genetic strategy used to target the mHb-IPN pathway (constitutive ChR2 expression in all ChAT+ and Tac1+ neurons) might not specifically target this projection. Future studies are needed to examine the precise mechanism whereby MOR signaling can potentiate glutamatergic neurotransmission in ChAT+ MHb-IPN projections."

---

## [Referee Report · Reviewer #3 (Public review)]

Summary:

Here the authors describe the role of mORs in synaptic glutamate release from substance P and cholinergic neurons in the medial habenula to interpeduncular nucleus (IPN) circuit in adult mice. They show that mOR activation reduces evoked glutamate release from substance P neurons yet increases evoked glutamate release and Ach release from cholinergic neurons. Unlike glutamate release, Ach release is only detected when potassium channels are blocked with 4-AP or dendrotoxin. The authors also report a previously unidentified glutamatergic input to IPR from SP neurons and describe the developmental timing of mOR- facilitation in adolescent mice.

Strengths:

- The experiments provide new insight into the role of mORs in controlling evoked glutamate release in a circuit with high levels of mORs and established roles in relevant behaviors.

- The experiments are rigorous, and the results are clear cut. The conclusions are supported by the data.

- The findings will be of interest to those working in the field of synaptic transmission and those interested in the function of the medial habenula or interpeduncular nucleus, as well as those seeking to understand the role of opioids on normal and pathological behaviors.

Weaknesses:

- The mechanistic underpinnings of these interesting and novel results are not pursued.

---

## [Author Response]

The following is the authors’ response to the original reviews.

**Reviewer #1 (Public Reviews):**
Weaknesses:Overall I find the data presented compelling, but I feel that the number of observations is quite low (typically n=3-7 neurons, typically one per animal). While I understand that only a few slices can be obtained for the IPN from each animal, the strength of the novel findings would be more convincing with more frequent observations (larger n, more than one per animal). The findings here suggest that the authors have identified a novel mechanism for the normal function of neurotransmission in the IPN, so it would be expected to be observable in almost any animal. Thus, it is not clear to me why the authors investigated so few neurons per slice and chose to combine different treatments into one group (e.g. Figure 2f), even if the treatments have the same expected effect.

This is a well taken suggestion. However, we must point out that we do perform statistical analyses on the original datasets and we believe that our conclusions are justified as acknowledged by the Reviewer. As the Reviewer is aware, the IPN is a small nucleus and with the slicing protocol used, we typically attain 1-2 slices per mouse that are suitable for recordings. Since most of the experiments in the manuscript deals with some form of pharmacological interrogation, we were reticent to use slices that are not naïve and therefore in general did not perform more than 1 cell recording per slice. Having said this, to comply with the Reviewer’s suggestion we have now performed additional experiments to increase the n number for certain experiments. We have amended all figures and legends to incorporate the additional data. We must point out that during the replotting of the data in the summary Figure 8i (previously Figure 7i) we noticed an error with the data representation of the TAC IPL data and have now corrected this oversight

Figure 2b,c.

500nM DAMGO effect on TAC IPL AMPAR EPSC – n increased from 5 to 9

Figure 3g.

500nM DAMGO effect on CHAT IPR AMPAR EPSC – n increased from 8 to 16 Effect of CTAP on DAMGO on CHAT IPR AMPAR EPSC – n increased from 4 to 7

Figure 3i.

500nm DAMGO or Met-enk effect in “silent” CHAT IPR AMPAR EPSC – n increased from 7 to 9

Figure 4e.

500nM DAMGO effect on ES coupling – Note: in the original version the n number was 5 and not 7 as written in the figure legend. We have now increased the n from 5 – 9.

Figure 5e,f.

500nM DAMGO effect on TAC IPR AMPAR EPSC – n increased from 5 to 9

Figure 7f.

Effect of DHβE on EPSC amplitude after application of DNQX/APV/4-AP or DTX-α – n increased from 7-9.

Figure 7g.

Emergence of nAChR EPSC after DTX – n increased from 4 to 7

Figure 7i.

Effect of ambenonium on nAChR amplitude and charge – n increased from 4 to 7

Supplementary Figure 3c and h

Effect of DAMGO after DNQX – n increased from 4 to 7

Effect of DNQX after DAMGO mediated potentiation – n increased from 3 to 5.

Throughout the study (Figs. 3i, 7f and 8h in the revised manuscript) we do indeed pool datasets that were amassed from different conditions since we were not directly investigating the possibility of any deviation in the extent of response between said treatments. For example, and as pointed out by the Reviewer, in Fig. 2F (now Fig. 3i) the use of DAMGO and met-ENK were merely employed to ascertain whether light-evoked synaptic transmission (ChATCre:ai32 mice) in cells that had no measurable EPSC could be pharmacologically “unsilenced” by mOR activation. Thus, the means by which mOR receptor was activated was not relevant to this specific question. Note: 2 more recordings are now added to this dataset (Fig. 3i) that were taken from ChATChR2/SSTCre:ai9 mice in response to the comment by this Reviewer below (“Are there baseline differences in the electrophysiological or morphological properties of these "silent" neurons compared to the responsive neurons?”). Similarly, in the revised Fig.7f we pooled data investigating the pharmacological block of the EPSC that emerged following application of either DNQX/APV/4-AP or DNQX/APV/DTX. Low concentrations 4-AP or DTX were interchangeably employed to reveal the DNQX-insensitive EPSC that we go on to show is indeed the nAChR response. Finally, in Fig. 8h, we pooled data demonstrating a lack of effect of DAMGO in potentiating both the glutamatergic and cholinergic arms of synaptic transmission in the OPRM1 KO mice. Again, here we were only interested in determining whether removal of mOR expression prevented potentiation of transmission mediated by mHB ChAT neurons irrespective of neurotransmitter modality. Thus, overall we were careful to only pool data in those instances where it would not change the interpretation and hence conclusions reached.

There are also significant sex differences in nAChR expression in the IPN that might not be functionally apparent using the low n presented here. It would be helpful to know which of the recorded neurons came from each sex, rather than presenting only the pooled data.

As the reviewer correctly states there are veins of literature concerning a divergence, based on sex, of not only nicotinic receptor expression but also behaviors associated with nicotine addiction. However, we have reanalyzed our datasets focusing on the extent of the mOR potentiation of glutamatergic and cholinergic transmission mediated by mHB ChAT neurons in IPR between male and female mice. Please refer to the Author response image 1 below. Although there is a possible trend towards a higher potentiation of nAChR in female mice, this was not found to be of statistical significance (see Author response image 1 below). We therefore chose not to split our data in the manuscript based on gender.

**Author response image 1. sa4fig1:** Comparison of the mOR (500nM DAMGO) mediated potentiation on evoked (a) AMPAR and (b) nAChR EPSCs in IPR between male and female mice.

There are also some particularly novel observations that are presented but not followed up on, and this creates a somewhat disjointed story. For example, in Figure 2, the authors identify neurons in which no response is elicited by light stimulation of ChAT-neurons, but the application of DAMGO (mOR agonist) un-silences these neurons. Are there baseline differences in the electrophysiological or morphological properties of these "silent" neurons compared to the responsive neurons?

Unfortunately, we did not routinely measure intrinsic properties of the recorded postsynaptic neurons nor systematically recovered biocytin fills to assess morphology. Therefore, it remains unclear whether the neurons in which there were none or minimal AMPAR-mediated EPSCs are distinct to the ones displaying measurable responses. The IPR is resident to GABAergic SST neurons that comprise the most numerous neuron type in this IPN subdivision. Although heavily outnumbered by the SST neurons there are additionally VGluT3+ glutamatergic neurons in IPN. The Reviewer is likely referring to a recent study investigating synaptic transmission specifically onto SST+ and VGluT3+ neurons in IPN demonstrating that mHB cholinergic mediated glutamatergic input is “weaker” onto the glutamatergic neurons. Furthermore, in some instances synaptic transmission onto this latter population can be “unsilenced” by GABAB receptor activation in a similar manner to that seen with mOR activation in this manuscript when IPR neurons are blindly targeted(Stinson & Ninan, 2025). Using a similar strategy as in this recent study(Stinson & Ninan, 2025), we now include experiments in which the ChATChR2 mouse was crossed with a SSTCre:Ai14. This allowed for recording of postsynaptic EPSCs in directly identified SST IPR neurons. We demonstrate that DAMGO can indeed increase glutamatergic EPSCs and in 2 of the cells where light activation demonstrated no appreciable AMPAR EPSC upon maximal LED light activation, DAMGO clearly “unsilenced” transmission. Thus, our additional analyses directly demonstrate that our original observations concerning mOR modulation extend to the mHb cholinergic AMPAR mediated input onto IPR SST neurons. This additional data is in the revised manuscript (Figure 3D-F, I). Future experimentation will be required to determine if the propensity of encountering a “silent” input that can be converted to robust synaptic transmission by mOR differs between these two cell types. Furthermore, it will be of interest to investigate if any differences exist in the magnitude of the cholinergic input or the mOR mediated potentiation of co-transmission between postsynaptic SST GABA and glutamatergic neuronal subtypes.

**Reviewer #2 (Public review)**
Weaknesses:The genetic strategy used to target the mHb-IPN pathway (constitutive expression in all ChAT+ and Tac1+ neurons) is not specific to this projection.

This is an important point made. We are acutely aware that the source of the synaptic input in IPN mediated by conditional expression of ChR2 employing using transgenic cre driver lines does not confer specificity to mHB. This is particularly relevant considering one of the novel observations here relates to a previously unidentified functional input from TAC1 neurons to the IPR. At this juncture we would like to point the Reviewer to the publicly available Connectivity Atlas provided by the Allen Brain Institute (https://connectivity.brain-map.org/). With reference to mHB TAC1 neuronal output, targeted viral injection into the habenula of Tac1Cre mice allows conditional expression of EGFP to SP neurons as evidenced by the predominant expression of reported fluorescence in dorsal mHB (see Author response image 2 a,b below). Tracing the axonal projections to the IPN clearly demonstrates dense fibers in IPL as expected but also arborization in IPR (Author response image 2 a,c) . This pattern is reminiscent of that seen in the transgenic Tac1Cre:ai9 or ai32 mice used in the current study (Figs. 1c, 2a, 5c). Closer inspection of the fibers in the IPR reveals putative synaptic bouton like structures as we have shown in Fig. 5a,b (Author response image 2 below).

**Author response image 2. sa4fig2:** Sterotaxic viral injection into mHB pf Tac1Cre mice taken from Allen Brain connectivity atlas (Link to Connectivity Atlas for mHb SP neuronal projection pattern).

These anatomical data suggest that part of the synaptic input to the IPR originates from mHB TAC1 neurons although we cannot fully discount additional synaptic input from other brain areas that may impinge on the IPR. Indeed, as the Reviewer points out, it is evident that other regions including the nucleus incertus send outputs to the IPN(Bueno et al., 2019; Liang et al., 2024; Lima et al., 2017). However, it is unclear if neuronal inputs from these alternate sources {Liang, 2024 #123;Lima, 2017 #33}{Bueno, 2019 #178} are glutamatergic in nature AND mediated by a TAC1/OPRM1-expressing neuronal population. Nevertheless, we have now modified text in the discussion to highlight the limitations of using a transgenic strategy.

In addition, a braking mechanism involving Kv1.2 has not been identified.

It is unclear to what the Reviewer is referring to here. Although most of our experiments pertaining to the brake on cholinergic transmission by potassium channels use low concentrations of 4-AP (50100μM) which have been used to block Shaker Kv1 channels there although at these concentrations there are additional action at other K+-channels such as Kv3, for instance. However, we essentially demonstrate that a selective Kv1.1 and Kv1.2 antagonist dendrotoxin replicates the 4-AP effects. We have now also included RNAseq data demonstrating the relative expression levels of Kv1 channel mRNA in mHb ChAT neurons (KCNA1 through KCNA6; Figure 6b). The complete absence of KCNA1 yet a high expression level of KCNA2 transcripts highly suggests a central role of Kv1.2 in unmasking nAChR mediated synaptic transmission.

**Reviewer #3 (Public review)**
Weaknesses:The significance of the ratio of AMPA versus nACh EPSCs shown in Figure 6 is unclear since nAChR EPSCs measured in the K+ channel blockers are compared to AMPA EPSCs in control (presumably 4-AP would also increase AMPA EPSCs).

We understand the Reviewer’s concern regarding the calculation of nicotinic/AMPA ratios since they are measured under differing conditions i.e. absence and presence of 4-AP, respectively. As the reviewer correctly points point 4-AP likely increases the amplitude of the AMPA receptor mediated EPSC. However, our intention of calculating this ratio was not to ascertain a measure of relative strengths of fast glutamatergic vs cholinergic transmission onto a given postsynaptic IPN neuron per se. Rather, we used the ratio as a means to normalize the size of the nicotinic receptor EPSC to the strength of the light stimulation (using the AMPA EPSC as the normalizing factor) in each individual recording. This permits a more meaningful comparison across cells/slices/mice . We apologize for the confusion and have amended the text in the results section to reflect this.

The mechanistic underpinnings of the most now results are not pursued. For example, the experiments do not provide new insight into the differential effects of evoked and spontaneous glutamate/Ach release by Gi/o coupled mORs, nor the differential threshold for glutamate versus Ach release.

Our major goal of the current manuscript was to provide a much-needed roadmap outlining the effects of opioids in the habenulo-interpeduncular axis. Of course, a full understanding of the mechanisms underlying such complex opioid actions at the molecular level will be of great value. We feel that this is beyond the scope of this already quite result dense manuscript but will be essential if directed manipulation of the circuit is to be leveraged to alter maladaptive behaviors associated with addiction/emotion during adolescence and in adult.

The authors note that blocking Kv1 channels typically enhances transmitter release by slowing action potential repolarization. The idea that Kv1 channels serve as a brake for Ach release in this system would be strengthened by showing that these channels are the target of neuromodulators or that they contribute to activity-dependent regulation that allows the brake to be released.

The exact mechanistic underpinnings that can potentially titer Kv1.2 availability and hence nAChR transmission would be essential to shed light on potential in vivo conditions under which this arm of neurotransmission can be modulated. However, we feel that detailed mechanistic interrogation constitutes significant work but one that future studies should aim to achieve. Thus, it presently remains unclear under what physiological or pathological scenarios result in attenuation of Kv1.2 to subsequently promote nAChR mediated transmission but as mentioned in the existing discussion future work to decipher such mechanisms would be of great value.

**Reviewer #1 (Recommendations for the authors):**
Overall I find this to be a very interesting and exciting paper, presenting novel findings that provide clarity for a problem that has persisted in the IPN field: that of the conundrum that light-evoked cholinergic signaling was challenging to observe despite the abundance of nAChRs in the IPN.Major concerns:(1) The n is quite low in most cases, and in many instances, data from one figure are replotted in another figure. Given that the findings presented here are expected in the normal condition, it should not be difficult to increase the n. A more robust number of observations would strengthen the novel findings presented here.

Please refer to the response to the public review above.

(2) In general, I find the organization of the figures somewhat disjointed. Sometimes it feels as if parts of the information presented in the results are split between figures, where it would make more sense to be together in a figure. For example, all the histology for each of the lines is in Figure 1, but only ephys data for one line is included there. It would be more logical to include the histology and ephys data for each line in its own figure. It would also be helpful to show the overlap of mOR expression with Tac1-Cre and ChAT-Cre terminals in the IPN. Likewise, the summarized Tac1Cre:Ai32 IPR data is in Figure 4, but the individual data is in Figure 5.

We introduce both ChAT and TAC1 cre lines in Figure 1 as an overview particularly for those readers who are not entirely familiar with the distinct afferent systems operating with the habenulointerpeduncular pathway. However, in compliance with the Reviewer’s suggestion we have now restructured the Figures. In the revised manuscript, the functional data pertaining to the various transmission modalities mediated by the distinct afferent systems impinging on the subdivision of the IPN tested are now split into their own dedicated figure as follows:

Figure 2.

mOR effect on TAC1neuronal glutamatergic output in IPL.

Figure 3.

mOR effect on CHAT neuronal glutamatergic output in IPR.

Figure 5.

mOR effect on TAC1neuronal glutamatergic output in IPR.

Figure 8.

mOR effect on CHAT neuronal cholinergic output in IPC.

Supp. Fig. 1 mOR effect on CHAT neuronal glutamatergic output in IPC.

We thank the Reviewer for their suggestions regarding the style of the manuscript. The restructuring has now resulted in a much better flow of the presented data.

(3) The discussion is largely satisfactory. However, a little more discussion of the integrative function of the IPN is warranted given the opposing effects of MOR activation in the Tac vs ChAT terminals, particularly in the context of both opioids and natural rewards.

We thank the reviewer for this comment. However, we feel the discussion is rather lengthy as is and therefore we refrained from including additional text.

Minor concerns:(1) The methods are missing key details. For example, the stock numbers of each of the strains of mice appear to have been left out. This is of particular importance for this paper as there are key differences between the ChAT-Cre lines that are available that would affect observed electrophysiological properties. As the authors indicate, the ChAT-ChR2 mice overexpress VAChT, while the ChAT-IRES-Cre mice do not have this problem. However, as presented it is unclear which mice are being used.

We apologize for the omission - the catalog numbers of the mice employed have now been included in the methods section.

We have now clearly included in each figure panel (single trace examples and pooled data) from which mice the data are taken from – in some instances the pooled data are from the two CHAT mouse strains employed. Despite the tendency of the ChATChR2 mice to demonstrate more pronounced nAChR mediated transmission (Fig. 7h), we justify pooling the data since we see no statistical significance in the effect of mOR activation on either potentiating AMPA or nAChR EPSCs (Please refer to response to Reviewer 2, Minor Concern point 2)

(2) Likewise, antibody dilutions used for staining are presented as both dilution and concentration, which is not typical.

We thank the reviewer for pointing out this inconsistency. We have amended the text in the methods to include only the working dilution for all antibodies employed in the study.

(3) There are minor typos throughout the manuscript.

All typos have been corrected.

**Reviewer #2 (Recommendations for the authors):**
The authors provide a thorough investigation into the subregion, and cell-type effect of mu opioid receptor (MOR) signaling on neurotransmission in the medial habenula to interpeduncular nucleus circuit (mHb-IPN). This circuit largely comprises two distinct populations of neurons: mHb substance P (Tac1+) and cholinergic (ChAT+) neurons. Corroborating prior work, the authors report that Tac1+ neurons preferentially innervate the lateral IPN (IPL) and rostral IPN (IPR), while ChAT+ neurons preferentially innervate the central IPN (IPC) and IPR. The densest expression of MOR is observed in the IPL and MOR agonists produce a canonical presynaptic depression of glutamatergic neurotransmission in this region. Interestingly, MOR signaling in the ChAT+ mHb projection to the IPR potentiates light-evoked glutamate and acetylcholine-mediated currents (EPSC), and this effect is mediated by a MOR-induced inhibition of Kv2.1 channels.Major concerns:(1) The method used for expressing channelrhodopsin (ChR2) into cholinergic and neurokinin neurons in the mHb (Ai32 mice crossed with Cre-driver lines) has limitations because all Tac1+/ChAT+ inputs to the IPN express ChR2 in this mouse. Importantly, the IPN receives inputs from multiple brain regions besides the IPN-containing neurons capable of releasing these neurotransmitters (PMID: 39270652). Thus, it would be important to isolate the contributions of the mHb-IPN pathway using virally expressed ChR2 in the mHb of Cre driver mice.

Please refer to the response to the public review above.

(2) Figure 4: The authors conclude that the sEPSC recorded from IPR originate from Tac1+ mHbIPR projections. However, this cannot be stated conclusively without additional experimentation. For instance, an optogenetic asynchronous release experiment. For these experiments it would also be important to express ChR2 virus in the mHb in Tac1- and ChAT-Cre mice since glutamate originating from other brain regions could contribute to a change in asynchronous EPSCs induced by DAMGO.

This is a well taken point. The incongruent effect of DAMGO on evoked CHAT neuronal EPSC amplitude and sEPSC frequency prompted us to consider the the possibility of differing effect of DAMGO on a secondary input. We agree that we do not show directly if the sEPSCs originate from a TAC1 neuronal population. Therefore, we have tempered our wording with regards the origin of the sEPSCs and have also restructured the Figure in question moving the sEPSC data into supplemental data (Figure 5 - supplement figure 1)

(3) Figure 5D: lt would be useful to provide a quantitative measure in a few mice of mOR fluorescence across development (e.g. integrated density of fluorescence in IPR).

We have now included mOR expression density across development (Fig. 6). Interestingly, the adult expression levels of mOR in the IPR are essentially reached at a very early developmental age (P10) yet we see stark differences in the role of mOR activation in modulating glutamatergic transmission mediated by mHB cholinergic neurons. Note: since we processed adult tissue (i.e. >p40) for these developmental analyses we utilized these slices to also include an analysis of the relative mOR expression density specifically in adults between the subdivisions of IPN in Fig. 1.

(4) Figure 6B: It would be useful to quantify the expression of Kcna2 in ChAT and Tac1 neurons (e.g. using FISH).

We thank the Reviewer for this suggestion. We have now included mRNA expression levels available from publicly available 10X RNA sequencing dataset provided by the Allen Brain Institute (Figure 7b).

(5) It would be informative to examine what the effects of MOR activation are on mHb projections to the (central) .

In response to this suggestion, we now have included additional data in the manuscript in putative IPC cells that clearly demonstrate a similar DAMGO elicited potentiation of AMPAR EPSC to that seen in IPR. These data are now included in the revised manuscript (Figure 3 - supplement figure 1; Fig. 8i).

(6) What is the proposed link between MOR activation and the inhibition of Kv1.2 (e.g. beta-Arrestin signaling, G beta-gamma interaction with Kv1.2, PKA inhibition?)

We apologize for any confusion. We do not directly test whether the potentiation of EPSCs upon mOR activation occurs via inhibition of Kv1.2.Although we have not directly tested this possibility we find it an unlikely underlying cellular mechanism, especially for the potentiation of the cholinergic arm of neurotransmission since in the presence of DNQX/APV, the activation of mOR does not result in any emergence of any nAChR EPSC (Figure 7 - supplemental figure 1a-c)

Minor concerns:(1) Methods: Jackson lab ID# for used mouse strains is missing.

We apologize for this omission and have now included the mouse strain catalog numbers.

(2) The authors use data from both ChAT-Cre x Ai32 and ChAT-ChR2 mice. It would be helpful to show some comparisons between the lines to justify merging data sets for some of the analyses as there appear to be differences between the lines (e.g. Figure 6G).

This is a well taken point. We have now provided a figure for the Reviewer (see Author response image 3 below) that illustrates the lack of significant difference between the mOR mediated potentiation of both mHB CHAT neuronal AMPAR and nAChR transmission between the two mouse lines employed despite a divergence in the extent of glutamatergic vs cholinergic transmission shown in Fig. 7g (previously Figure 6g). We have chosen not to include this data in the revised manuscript.

**Author response image 3. sa4fig3:** Comparison of the mOR (500nM DAMGO) mediated potentiation on evoked AMPAR **(a)** and nAChR **(b)**EPSCs in IPR between ChATCre:Ai32 and ChATChR2 mice.

(3) Line 154: How was it determined that the EPSC is glutamatergic?

We apologize for any confusion. In the revised manuscript we now clearly point to the relevant figures (see Figure 5 supplement figure 1a and Figure 7 supplement figure 1a - e) in the Results section where we determine that both the sEPSCs and ChAT mediated light evoked EPSCs recorded under baseline conditions are totally blocked by DNQX and hence are exclusively AMPAR events

(4) It would be helpful to discuss the differences between GABA-B mediated potentiation of mHbIPN signaling and the current data in more detail.

We are unclear as to what differences the Reviewer is referring to. At least from the perspective of ChAT neuronal mediated synaptic transmission, other groups (and in the current study; **Fig. 7h**) have clearly shown that GABA_B_ activation markedly potentiates synaptic transmission like mOR activation. Nevertheless, based on our novel findings it would be of interest to determine whether the influence of GABA_B_ is inhibitory onto the TAC mediated input in IPR and whether there is a developmental regulation of this effect as we demonstrate upon mOR activation. These additional comparisons between the effect of the two Gi-linked receptors may shed light onto the similarity, or lack thereof, regarding the underlying cellular mechanisms. We now have included a few sentences in the discussion to highlight this (pg 11, para 1).

**Reviewer #3 (Recommendations for the authors):**
The abstract was confusing at first read due to the complex language, particularly the sentence starting with... Further, specific potassium channels...The authors might want to consider simplifying the description of the experiments and the results to clarify the content of the manuscript for readers who many only read the abstract.

We have altered the wording of the abstract and hope it is now more reader friendly.

The opposite effect of mOR activation on spontaneous EPSCs versus electrical or ChR2-evoked EPSCs is very interesting and raises the issue of which measure is most physiologically relevant. For example, it is unclear whether sEPSCs arise primarily from cholinergic neurons (that are spontaneously active in the slice, Figure 3), and if so, does mOR activation suppress or enhance cholinergic neuron excitability and/or recruitment by ChR2? While a full analysis of this question is beyond the scope of this manuscript, the assumption that glutamate release assayed by electrical/ChR2 evoked transmission is the most physiologically relevant might merit some discussion since sEPSCs presumably also reflect action-potential dependent glutamate release. One wonders whether mORs hyperpolarize cholinergic neurons to reduce spontaneous spiking yet enhance fiber recruitment by ChR2 or an electrical stimulus (i.e. by removing Na channel inactivation). The authors have clearly stated that they do not know where the mORs are located, and that the effects arising from disinhibition are likely complex. But they also might discuss whether glutamate release following synchronous activation of a fiber pathway by ChR2 or electrode is more or less physiologically relevant than glutamate release assayed during spontaneous activity. It seems likely that an equivalent experiment to Figure 3D, E using spontaneous spiking of IPR neurons would show that spiking is reduced by mOR activation.

We thank the Reviewer for this comment. As pointed it would be of interest to dissect the “network” effect of mOR activation but as the Reviewer acknowledges this is beyond the scope of the current manuscript. The Reviewer is correct in postulating that mOR activation results in hyperpolarization of mHB ChAT neurons. A recent study(Singhal et al 2025) demonstrate that a subpopulation of ChAT neurons undergoes a reduction in firing frequency following DAMGO application. This is corroborated by our own observations although we chose not to include this data in our current manuscript.

Additionally, the Reviewer questions whether ChR2/electrical stimulation is physiological. This is a well taken point and of course the simultaneous activation of potentially all possible axonal release sites is not the mode under which the circuit operates. Nevertheless, our data clearly demonstrates the ability of mORs to modulate release under these circumstances that must reflect an impact on spontaneous action potential driven evoked release. Although the suggested experiment could shed light on the synaptic outcomes of mOR receptor activation on ES coupling of downstream IPN neurons. Interpretation of the outcome would be confounded by the fact that postsynaptic IPN neurons also express mORs . Thus, we would not be able to isolate the effects of presynaptic changes in modulating ES coupling from any direct postsynaptic effect on the recorded cell when in current clamp.

Together these additional sites of action of mOR (i.e. mHB ChAT somatodendritic and postsynaptic IPN neuron) only serve to further highlight the complex nature of the actions of opioids on the habenulo-interpeduncular axis warranting future work to fully understand the physiological and pathological effects on the habenulo-interpeduncular axis as a whole.

The idea that Kv2.1 channels serve as a brake raises the question of whether they contribute to activity-dependent action potential broadening to facilitate Ach release during trains of stimuli.

This is an interesting suggestion and one that we had considered ourselves. Indeed, as the Reviewer is likely aware and as mentioned in the manuscript, previous studies have shown nAChR signaling can be revealed under conditions of multiple stimulations given at relatively high frequencies. We therefore attempted to perform high frequency stimulation (20 stimulations at 25Hz and 50Hz) in the presence of ionotropic glutamatergic receptor antagonists DNQX and APV. We have now included this data in the revised manuscript (Figure 7 supplement figure 1d,e). As shown, this failed to engage nAChR mediated synaptic transmission in our hands. Interestingly there is evidence from reduced expression systems demonstrating that Kv1.2 channels undergo use-dependent potentiation(Baronas et al., 2015) in contrast to that seen with other K+-channels. Whether this is the case for the axonal Kv1.2 channels on mHB axonal terminals in situ is not known but this may explain the inability to reveal nAChR EPSCs upon delivery of such stimulation paradigms.

References

Baronas, V. A., McGuinness, B. R., Brigidi, G. S., Gomm Kolisko, R. N., Vilin, Y. Y., Kim, R. Y., … Kurata, H. T. (2015). Use-dependent activation of neuronal Kv1.2 channel complexes. J Neurosci, 35(8), 3515-3524. doi:10.1523/JNEUROSCI.4518-13.2015

Bueno, D., Lima, L. B., Souza, R., Goncalves, L., Leite, F., Souza, S., … Metzger, M. (2019). Connections of the laterodorsal tegmental nucleus with the habenular-interpeduncular-raphe system. J Comp Neurol, 527(18), 3046-3072. doi:10.1002/cne.24729

Liang, J., Zhou, Y., Feng, Q., Zhou, Y., Jiang, T., Ren, M., … Luo, M. (2024). A brainstem circuit amplifies aversion. Neuron. doi:10.1016/j.neuron.2024.08.010

Lima, L. B., Bueno, D., Leite, F., Souza, S., Goncalves, L., Furigo, I. C., … Metzger, M. (2017). Afferent and efferent connections of the interpeduncular nucleus with special reference to circuits involving the habenula and raphe nuclei. J Comp Neurol, 525(10), 2411-2442. doi:10.1002/cne.24217

Singhal, S. M., Szlaga, A., Chen, Y. C., Conrad, W. S., & Hnasko, T. S. (2025). Mu-opioid receptor activation potentiates excitatory transmission at the habenulo-peduncular synapse. Cell Rep, 44(7), 115874. doi:10.1016/j.celrep.2025.115874

Stinson, H.E., & Ninan, I. (2025). GABA(B) receptor-mediated potentiation of ventral medial habenula glutamatergic transmission in GABAergic and glutamatergic interpeduncular nucleus neurons. bioRxiv doi.10.1101/2025.01.03.631193